# Mining the heparinome for cryptic antimicrobial peptides that selectively kill Gram-negative bacteria

Roberto Bello-Madruga [1,7], Daniel Sandín[1,7], Javier Valle[2], Jordi Gómez [1], Laura Comas[3], María Nieves Larrosa [4,5,6], Juan José González-López[4,5,6], María Ángeles Jiménez [3], David Andreu [2✉] & Marc Torrent [1✉]

## Abstract

**Glycosaminoglycan (GAG)-binding proteins regulating essential processes such as cell growth and migration are essential for cell homeostasis. As both GAGs and the lipid A disaccharide core of Gram-negative bacteria contain negatively charged disaccharide units, we hypothesized that GAG-binding proteins could also recognize LPS and enclose cryptic antibiotic motifs. Here, we report novel antimicrobial peptides (AMPs) derived from heparin-binding proteins (HBPs), with specific activity against Gram-negative bacteria and high LPS binding. We used computational tools to locate antimicrobial regions in 82% of HBPs, most of those colocalizing with putative heparin-binding sites. To validate these results, we synthesized five candidates [HBP-1-5] that showed remarkable activity against Gram-negative bacteria, as well as a strong correlation between heparin and LPS binding. Structural characterization of these AMPs shows that heparin or LPS recognition promotes a conformational arrangement that favors binding. Among all analogs, HBP-5 displayed the highest affinity for both heparin and LPS, with antimicrobial activities against Gram-negative bacteria at the nanomolar range. These results suggest that GAG-binding proteins are involved in LPS recognition, which allows them to act also as antimicrobial proteins. Some of the peptides reported here, particularly HBP-5, constitute a new class of AMPs with specific activity against Gram-negative bacteria.**

**Keywords** Antimicrobial Peptide; Heparin-binding Protein; Lipopolysaccharide; Heparin; Glycosaminoglycans
**Subject Category** Microbiology, Virology & Host Pathogen Interaction

## Introduction

Glycosaminoglycan (GAG)-binding proteins are a heterogeneous group of proteins mostly associated with the cell surface and the extracellular matrix (Pomin and Mulloy, 2018; Shi et al, 2021). They mediate a plethora of functions including signaling, cell proliferation, and coagulation (Gulati and Poluri, 2016; Saied-Santiago and Bülow, 2018). Up to date, most studies of the GAG interactome have focused on protein interactions with heparin, a highly sulfated form of heparan sulfate, due to the commercial availability of heparin and heparin-Sepharose (Vallet et al, 2021). This has allowed defining the heparin interactome, a highly interconnected network of proteins functionally linked to physiological and pathological processes (Ori et al, 2011). Although the structural nature of these proteins is diverse, they share common features, such as the presence of certain domains and motifs (Peysselon and Ricard-Blum, 2014; Weiss et al, 2017). In particular, the CPC' clip motif is the major contributor to the attachment of heparin (and other sulfated GAGs) to GAG-binding proteins (Iannuzzi et al, 2015). The motif involves two cationic (Arg or Lys) and one polar (Asn, Gln, Thr, Tyr or Ser, more rarely Arg or Lys) residues with conserved distances between the α carbons and the side-chain center of gravity, defining a clip-like structure where heparin is lodged (Torrent et al, 2012b). The CPC' clip motif is conserved among all HBPs deposited in the PDB and can be found in many proteins with reported heparin-binding capacity (Torrent et al, 2012b).

Recently, we showed that negatively charged polysaccharide-containing polymers, such as heparin and lipopolysaccharides (LPS), can compete for similar binding sites in peptides, and that the CPC' clip motif is essential to bind both ligands (Pulido et al, 2017). Our results provide a structural framework to explain why these polymers can cross-interact with the same proteins and peptides and thus contribute to the regulation of apparently unrelated processes in the body. A paradigmatic example is FhuA, an *E. coli* transmembrane protein involved in the transport of antibiotics such as albomycin and rifamycin (Braun, 2009). FhuA can bind glucosamine phosphate groups in LPS (Shearer et al, 2019), and we confirmed that a short peptide (YI12WF) retaining most of the LPS-binding affinity of the original protein can also

[1]The Systems Biology of Infection Laboratory, Department of Biochemistry and Molecular Biology, Biosciences Faculty, Universitat Autònoma de Barcelona, Cerdanyola del Vallès 08193, Spain. [2]Department of Medicine and Life Sciences, Universitat Pompeu Fabra, Barcelona Biomedical Research Park, Barcelona 08003, Spain. [3]Departamento de Química-Física Biológica, Instituto de Química Física Blas Cabrera (IQF-CSIC). Serrano 119, Madrid 28006, Spain. [4]Servei de Microbiologia, Hospital Universitari Vall d'Hebron, Barcelona 08035, Spain. [5]Departament de Genètica i Microbiologia, Universitat Autònoma de Barcelona, Cerdanyola del Vallès 08193, Spain. [6]CIBER de Enfermedades Infecciosas (CIBERINFEC), Instituto de Salud Carlos III, Madrid, Spain. [7]These authors contributed equally: Roberto Bello-Madruga, Daniel Sandín. ✉E-mail: david.andreu@upf.edu; marc.torrent@uab.cat

bind heparin with high affinity (Bhunia et al, 2009; Pulido et al, 2017). When the CPC' residues in these peptides are mutated, heparin and LPS-binding activities are largely lost, proving the motif as essential for both ligands. Heinzelmann and Bosshart also showed that human lipopolysaccharide-binding protein (hLBP) can bind heparin and enhance the pro-inflammatory responses to LPS of blood monocytes (Li et al, 2011). Again, the crystal structure of hLBP bound to N-acetyl-D-glucosamine shows a CPC' clip motif that could potentially bind heparin. Such observations may prove generalizable to other LPS-binding proteins and may reveal a biological interplay between LPS and heparin. Whether the reverse is true—i.e., HBPs playing a role in LPS binding and potentially in antimicrobial activity—is currently unknown.

Here we show that HBPs contain cryptic AMPs that overlap with heparin-binding regions containing a CPC' motif. These AMPs show strong selective antimicrobial activity for Gram-negative bacteria. They also bind heparin and LPS with high affinity and disrupt the bacterial cell wall. Our results suggest that LPS and heparin bind similar regions in proteins, provided they contain a CPC' clip motif. HBPs therefore represent a source for new antimicrobials effective against antibiotic-resistant pathogens.

# Results

## Linking heparin affinity and antimicrobial activity

Despite the differences between GAGs and LPS, both contain negatively charged disaccharides in their structure. GAGs are polymers based on variably sulfated repeating disaccharide units. For example, the most common form of heparin is a sulfated disaccharide composed of iduronic acid and glucosamine linked through a $\beta$ $(1 \rightarrow 4)$ bond (IdoA(2S)-GlcNS(6S); Fig. 1A). For its part, LPS is composed of a polysaccharide antigen linked to a lipid A molecule, which is, in turn, a phosphorylated glucosamine (GlcN) disaccharide decorated with multiple fatty acids. The two GlcN units are linked by a $\beta$ $(1 \rightarrow 6)$ bond, and normally contain one phosphate group each (Fig. 1B). Based on these structural similarities, we hypothesized that HBPs could also potentially bind the phosphorylated GlcN units of LPS. As heparin-binding sites are commonly associated with short sequential motifs, we reasoned that specific short regions in HBPs could behave as AMPs, binding first to LPS and later destabilizing the outer cell wall and the bacterial membranes.

To validate our hypothesis, we inspected all reported HBPs (Appendix File 1) using the AntiMicrobial Peptide Analyzer (AMPA), a prediction algorithm that can detect the presence of cryptic antimicrobial segments in proteins (Ramos-Llorens et al, 2024; Torrent et al, 2012a). Using the default parameters, AMPA detected potential antimicrobial regions in 82% of the HBP set, suggesting that most HBPs contain cryptic AMPs that can be mined by AMPA. According to our hypothesis, these regions should colocalize with heparin-binding sites in HBPs. To ascertain whether the AMPA-retrieved cryptic AMPs could indeed bind GAGs, we first resorted to molecular docking. In AutoDock Vina, a docking region (grid) centered on the antimicrobial segment detected by AMPA was defined and docked with a heparin disaccharide I–S (H1S, α-ΔUA-2S-[1 → 4]-GlcNS-6S). Results show 76% of the cryptic antimicrobial regions as potential binders of H1S, with

affinity comparable to well-defined heparin-binding motifs (Fig. 1C). We also examined the presence of CPC' clips in HBPs with a docking score higher than the average energy calculated for experimentally validated HBPs (−6.8 kcal/mol, 30 proteins) and found that 74% of such regions contain a CPC' motif with geometric distances compatible with GAG anchoring (Fig. 1D). We therefore concluded that heparin-binding regions significantly overlap with cryptic antimicrobial regions in HBPs, hence structural co-localization of antimicrobial activity and GAG recognition can be posited.

## Validation of cryptic AMPs by synthetic HBPs

To confirm our hypothesis, we synthesized five peptides (Table 1) reproducing regions with the highest AMPA score that also contained a CPC' clip motif and exhibited an amino acid distribution that creates cationic patches surrounded by hydrophobic residues (Appendix Table S1; Fig. EV1; Appendix Fig. S1). We first used affinity chromatography to check whether the peptides were able to bind heparin, hence that the binding region had been successfully delimited, and indeed found the retention times for all peptides to be higher than control antimicrobial peptide LL-37 (Table 1). In two cases, **HBP-4** and **HBP-5**, the affinity was so high that up to 98% buffer B had to be used to dislodge them from the heparin column. So, we could safely conclude that all peptides showed medium-to-strong heparin binding, evoking that of parental HBPs.

We next inspected antimicrobial activity. Minimal inhibitory concentration (MIC) and minimal bactericidal concentration (MBC) were determined on a panel of Gram-negative and Gram-positive bacteria. The five peptides displayed strong activity against Gram-negative (*Escherichia coli*, *Acinetobacter baumannii*, and *Pseudomonas aeruginosa*) while being much less active against Gram-positive bacteria (*Staphylococcus aureus*, *Enterococcus faecium*, and *Listeria monocytogenes;* Table 2). This observation is consistent with our hypothesis that, lacking LPS, Gram-positives are much less susceptible than Gram-negatives to AMPs. Also, in tune with the hypothesis, peptides with the strongest heparin affinity (**HBP-4** and **HBP-5)** had the best antimicrobial activity, correlating both observations. In contrast **HBP-2**, the peptide with the lowest affinity, did not show any significant difference in activity between Gram-positive and Gram-negative bacteria, except for *S. aureus*. Antimicrobial activity was also retained against clinical isolates of Gram-negative strains (Appendix Table S2). Specifically, **HBP-4** and **HBP-5** were remarkably active, including multidrug-resistant *P. aeruginosa* strains. Given these encouraging results, we inspected the hemolytic capacity of the peptides as a benchmark of their therapeutic potential as antimicrobials (Appendix Table S3). Erythrocyte lysis was low for all peptides; reaching only 15% at 125 μM, in contrast to >30% lysis for LL-37 at the same concentration. On mammalian (MRC-5 and HepG2) cells, similarly favorable results were again found. For **HBP-4**, the (relatively) more cytotoxic peptide, $LC_{50}$ was comparable to LL-37, but **HBP-5** was significantly better. Interestingly, another peptide called NLF20, isolated from the same region of heparin cofactor 2, also displays strong antimicrobial properties (Kalle et al, 2013; Kalle et al, 2014; Papareddy et al, 2010). Overall, **HBP-5** emerges as the most attractive analog, with a selectivity ratio ($LC_{50}$/MIC) between 50 and 800 (depending

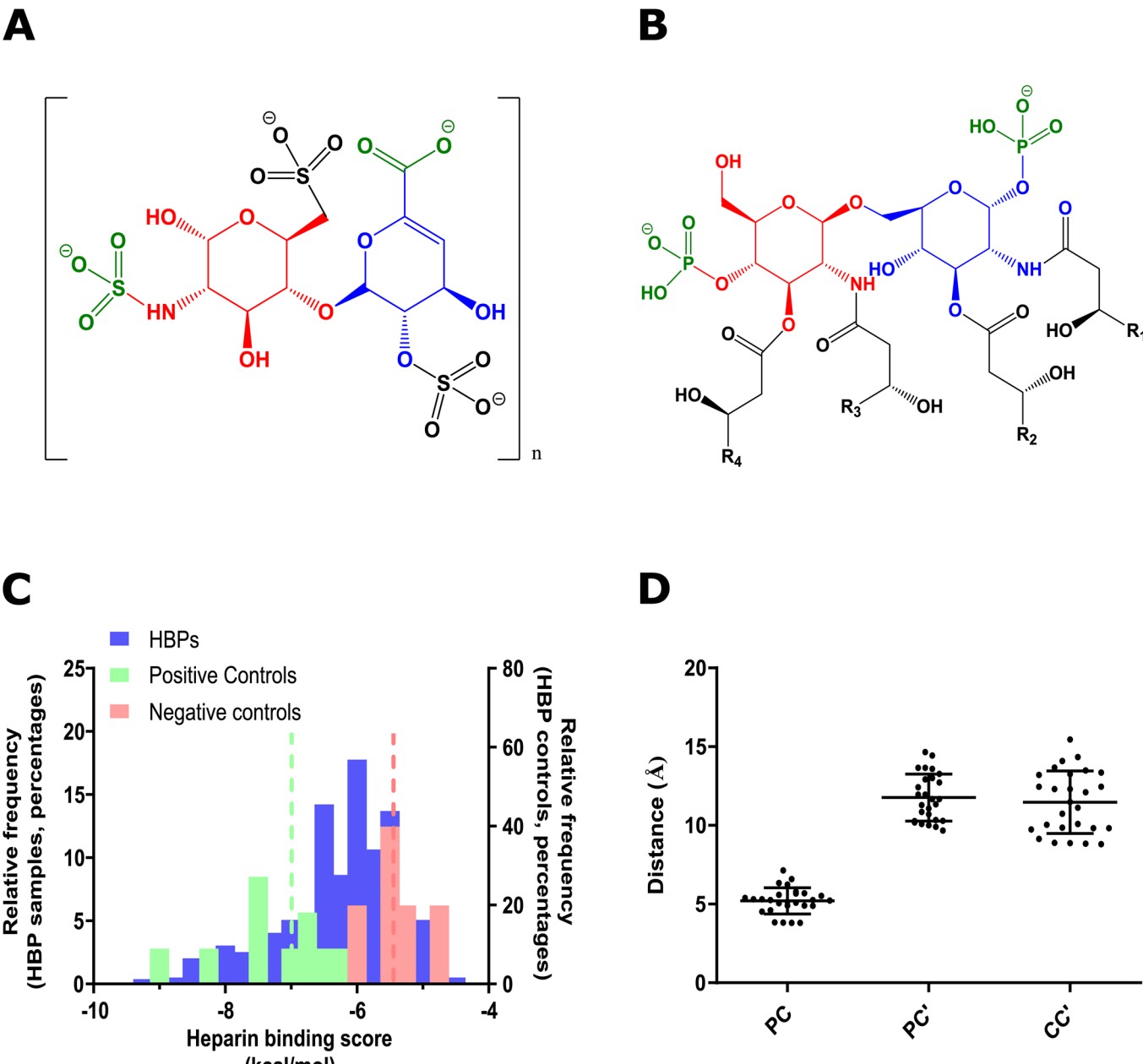

**Figure 1. Structural and bioinformatics analysis of HBPs.**

Structure of (**A**) heparin disaccharide and (**B**) lipid A disaccharide central axis. Similar regions in both structures are highlighted with colors. (**C**) Affinity score distribution of AMPs (blue), positive controls (green, dotted line in green refers to their mean, $-7.0 \pm 1.1$ kcal/mol), and negative controls (red, dotted line in red refers to their mean, $-5.4 \pm 0.5$ kcal/mol). (**D**) Distances between cationic and polar residues in the best candidates ($n = 27$) with CPC' motifs detected. Error bars correspond to the standard deviation of the mean. Reference values for PC, PC' and CC' residues in CPC' motifs are $6.0 \pm 1.9$ Å for PC, $11.6 \pm 1.6$ Å for PC' and $11.4 \pm 2.4$ Å for CC' [16]. Source data are available online for this figure.

on bacterial strain) that must be regarded as outstanding for AMPs, and that confirms the hypothesis that HBPs contain cryptic AMPs.

## Mechanism of action

Given the interesting antimicrobial profiles of HBPs, we investigated their mechanism of action to determine if activity could be related to the interaction with LPS, hence with the cell wall. First, we analyzed LPS-binding affinity with the BODIPY-cadaverine assay. **HBP-4** and **HBP-5**, the peptides with the best antimicrobial activity, also exhibited the strongest LPS binding, exceeding that of LL-37, while the other analogs showed moderate binding, **HBP-2** being the poorest one, again in tune with low antimicrobial activity (Fig. 2A; Appendix Table S4). Consistently, peptide NLF20 was also shown to be able to disrupt lipopolysaccharide aggregates

**Table 1. Synthetic peptide analytical data and heparin affinities.**

| Peptide | Sequence | HPLC retention time (min)[a] | Molecular mass (Da) | | Heparin affinity (% elution buffer)[b] |
|---|---|---|---|---|---|
| | | | Theory | Found | |
| HBP-1 | RWHLTHRPKTGYIRVLVH-amide | 1.5 | 2269.7 | 2268.7 | 60 |
| HBP-2 | RFYLSKKKWVMVP-amide | 1.7 | 1682.1 | 1681.1 | 59 |
| HBP-3 | FRFKRKLPKYLLF-amide | 1.9 | 1756.2 | 1755.2 | 68 |
| HBP-4 | GWKDKKSYRWFLQHRPQVGYIRVRFY-amide | 1.9 | 3414.9 | 3414.0 | 82 |
| HBP-5 | HNLFRKLTHRLFRRNFGYTLRSV-amide | 1.9 | 2932.4 | 2931.4 | 98 |
| LL-37 | LLGDFFRKSKEKIGKEFKRIVQRIKDFLRNLVPRTES-amide | 2.1 | 4493.3 | 4492.3 | 50 |

[a]10 to 60% of solvent B (ACN with 0.036% TFA) into solvent A ($H_2O$ with 0.045% TFA) in a 3-minute run.
[b]Elution buffer was 2 M NaCl in 10 mM $Na_2HPO_4$.

**Table 2. MIC and MBC data for all peptides against reference strains.**

| | MIC/MBC (µM) | | | | | |
|---|---|---|---|---|---|---|
| | *E. coli* | *A. baumannii* | *Pseudomonas sp* | *S. aureus* | *E. faecium* | *L. monocytogenes* |
| HBP-1 | 1.6/1.6 | 1.6/1.6 | 3.1/3.1 | 50/ > 100 | ≥100/ > 100 | 6.3/12.5 |
| HBP-2 | 12.5/25 | 50/50 | 25/50 | >100/ > 100 | >100/ > 100 | 37.5/ > 100 |
| HBP-3 | 3.1/3.1 | 0.8/0.8 | 6.3/6.3 | 25/50 | >100/ > 100 | 6.3/12.5 |
| HBP-4 | 0.2/0.2 | 0.8/0.8 | 0.8/0.8 | 3.1/12.5 | 12.5/50 | 1.6/3.1 |
| HBP-5 | 0.4/0.4 | 0.2/0.2 | 0.8/0.8 | 6.3/6.3 | 3.1/6.3 | 1.6/3.1 |

(Singh et al, 2013). This correlation between heparin and LPS affinities strongly suggests that both activities are related (Fig. EV2). The results are also consistent with the lethality curves measured in *E. coli*, in which **HBP-4** and **HBP-5** are fast acting, far more than LL-37, while **HBP-1** and **HBP-2** are the slowest ones (Fig. 2B). All peptides showed membrane depolarization abilities comparable to LL-37, according to the DiSC3(5) assay (Fig. 2C; Appendix Table S4), with **HBP-5** again scoring highest and **HBP-2** lowest among all analogs.

We next proceeded to evaluate the outer membrane (OM) permeability of HBPs in the 1-N-phenylnaphthylamine (NPN) assay. NPN is a small hydrophobic molecule unable to cross the bacterial OM. However, when the OM is damaged, NPN can interact with lipids and proteins inside the cell, with an ensuing increase in fluorescence that effectively acts as a reporter for OM permeability. All five peptides caused OM damage, with a remarkable increase in fluorescence intensity observed upon 4 h incubation (Fig. 2D). As in the depolarization assay, peptides with stronger antibacterial activity against Gram-negative bacteria displayed also increased OM permeabilization. This effect was especially clear for **HBP-4** and **HBP-5**, which reached higher fluorescence intensities than control LL-37. Finally, to directly detect cell wall damage, the morphology of peptide-incubated *E. coli* cells was observed by scanning electron microscopy, which showed a clear disruption of the bacterial envelope in all cases (Fig. 2E).

Taken together, the above results confirm that HBPs have the ability to bind LPS, compromise OM integrity, disrupt cell structure and promote depolarization, eventually triggering bacterial cell death.

## Structural characterization

To investigate any structural changes occurring upon interaction of HBPs with heparin or cell membranes, we obtained circular dichroism (CD) spectra in buffer, SDS, LPS, and heparin (Fig. 3A; Appendix Tables S5–8). In almost all cases, the structures in water were random, with minima at ~200 nm. In the presence of SDS micelles, **HBP-3** and **HBP-5** displayed minima near 208 and 222 nm, with a positive band at ~190 nm, evidencing a shift towards α-helical conformation (Avitabile et al, 2014). For peptides **HBP-1**, **HBP-2**, and **HBP-4**, a shift towards a minimum at 218 nm was observed, suggesting a β-strand structure (Avitabile et al, 2014). This behavior is typical of AMPs; the random-to-structure transition favors partial insertion into the membrane, promoting depolarization (Bello-Madruga and Torrent Burgas, 2024). With LPS, again a transition from random to either α-helix or β-strand was observed for **HBP-4** and **HBP-5**, less pronounced for the other analogs. This behavior was repeated for all peptides in the presence of heparin, except for **HBP-2**, which remained in a disordered conformation. These results are consistent with the above antimicrobial and LPS-binding assays in suggesting that LPS and heparin binding trigger a structural arrangement into a more defined, antimicrobially effective structure which, in all cases, is similar to that adopted by the peptide segment in the corresponding original protein (Fig. 3A).

As **HBP-5** was the most interesting analog in terms of antimicrobial activity and heparin binding, we decided to inspect its solution structure by NMR in (i) water, (ii) DPC micelles, and (iii) in the presence of heparin analogs. After assigning ¹H and ¹³C chemical shifts, we performed a qualitative analysis of the $\Delta\delta_{H\alpha}$ and

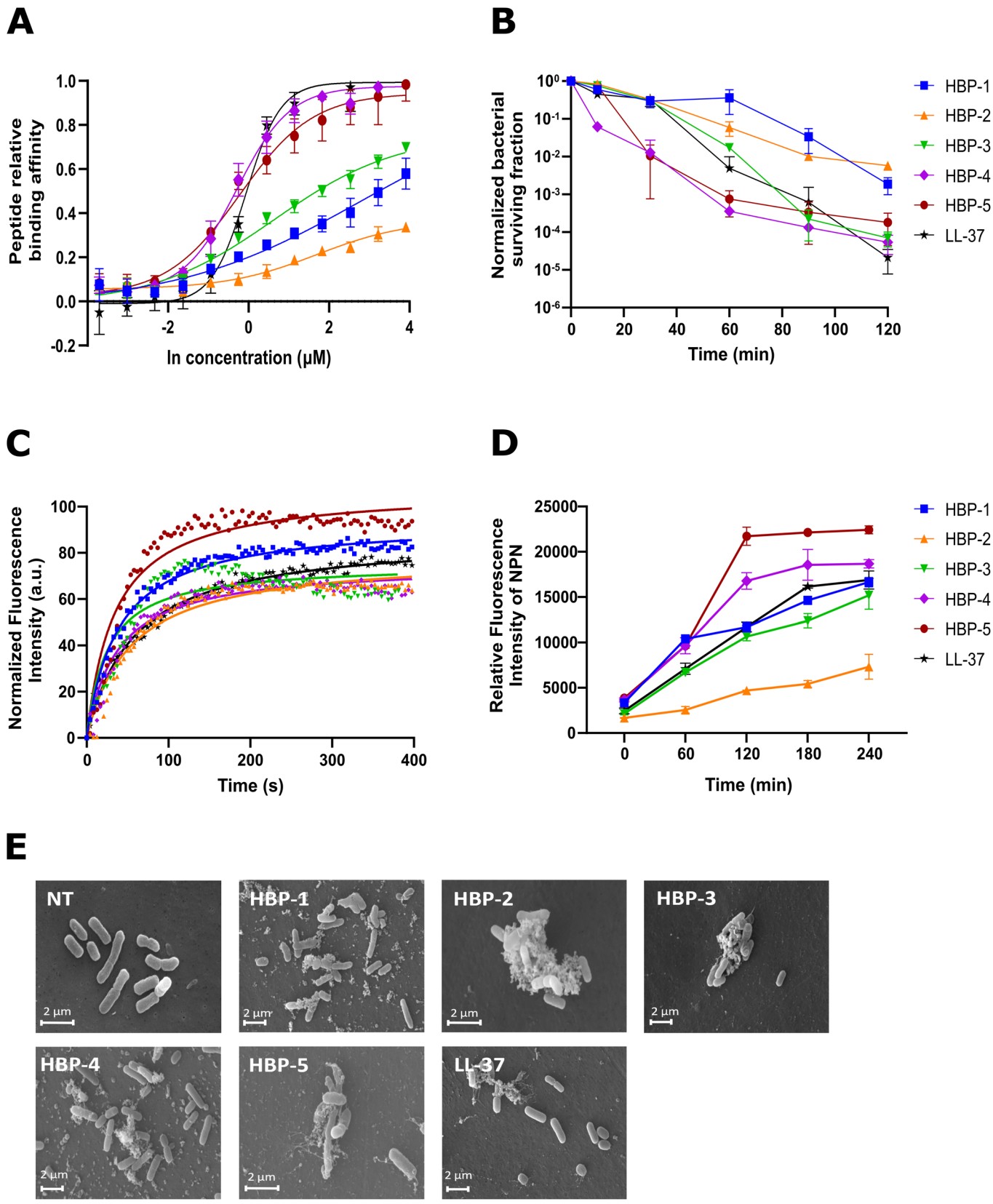

**Figure 2. Mechanism of action of HBP-derived antimicrobial peptides.**

(A) LPS affinity measured as an increase in fluorescence emission ($\lambda_{em} = 620$ nm) of BODIPY-cadaverine at different peptide concentrations. (B) Bactericidal activity kinetics obtained by treating *E. coli* planktonic cultures with HBPs. Peptide concentrations were 50 µM HBP-1, 150 µM HBP-2, 50 µM HBP-3, 25 µM HBP-4, 6.3 µM HBP-5 and 50 µM LL-37. (C) Cell depolarization measured as DiSC3(5) fluorescence emission increase after incubating *E. coli* with HBPs. LL-37 was used as a positive control of membrane depolarization. All peptides were tested at 10 µM, except HBP-2 at 20 µM. (D) Outer membrane permeabilization of HBPs peptides measured by detecting the fluorescence intensity of NPN (after normalization) in *E. coli* BW25113. (E) SEM pictures of *E. coli* cells treated with HBPs at the same concentrations used in (C), after 2 h incubation at 37 °C. NT = non peptide control. In plots (A, B, D), data are shown as the mean ± SEM of the three independent experiments. Source data are available online for this figure.

$\Delta\delta_{C\alpha}$ conformational shifts ($\Delta\delta = \delta^{observed} - \delta^{random\ coil}$, ppm; see Methods and Fig. EV3). The fact that $\Delta\delta_{H\alpha}$ and $\Delta\delta_{C\alpha}$ values are within the random coil range indicates that **HBP-5** is mainly disordered in aqueous solution, as observed previously by CD. In DPC micelles, the stretches of negative $\Delta\delta_{H\alpha}$ and positive $\Delta\delta_{C\alpha}$ values indicate the presence of a highly populated helix structure spanning approximately residues 3-11 (Appendix Table S9). Structure calculation, which includes medium and long-range distance restraints derived from the observed NOE cross-peaks (see "Methods" and Appendix Table S10), showed a well-defined N-terminal helix spanning residues 3–15, three residues longer than deduced from qualitative analysis of $\Delta\delta_{Ha}$ and $\Delta\delta_{Ca}$ values, and a less ordered non-regular turn-like motif involving residues 18–22 (Fig. 3B). The relative arrangement of the α-helix and the turn-like motif is poorly defined. Unfortunately, attempts to retrieve an NMR 3D structure of **HBP-5** complexed with heparin analogs were unsuccessful. Spectra of **HBP-5** with either the fondaparinux (Arixtra®) pentasaccharide or the simpler H1S disaccharide acting as heparin analogs did not provide any NOE cross-peaks evidencing intermolecular peptide-sugar contacts, mostly due to the substantial sample precipitation observed, particularly for fondaparinux. In the presence of an equimolar amount of H1S disaccharide, many cross-peaks are shifted relative to free **HBP-5** (see Appendix Fig. S2). To identify which residues are most affected upon disaccharide interaction, weighted chemical shift differences ($\Delta\delta_w$, ppm) were plotted as a function of peptide sequence (Fig. 3C). It is clear in this plot that significant differences are mainly located at the central section, residues 8–14 (Fig. 3C). In view of this, and to obtain additional insights into heparin binding of **HBP-5**, we performed a molecular dynamics simulation with fondaparinux. The results show that the pentasaccharide remains in contact with the peptide all along the simulation time, suggesting strong binding. Specifically, we observed a persistent salt bridge between the Arg13 side chain and the S6 sulfate group of the pentasaccharide (Fig. 3D). Another salt bridge between Arg10 and the S6 sulfate, plus a loosely defined hydrogen bond between His9 and the S3 sulfate were also identified. These three residues (His9, Arg10, and Arg14) form a CPC' motif with their relative distances maintained throughout the simulation (Fig. 3E; Appendix Fig. S3), altogether suggesting a CPC' clip as a relevant binding element.

## Insights into the CPC' motif of HBP-5 and its implication on the antibacterial mechanism

To assess the significance of the CPC' motif in HBP-5, two peptides, **R10** and **R14** (Appendix Table S11), with the cationic residues within the CPC' clip mutated to glutamine, were made, as well as **ΔCPC'** (Appendix Table S11), a peptide where the three residues defining the CPC' motif were replaced by alanine. For mutants **R10** and **R14** antibacterial activity was similar to that of **HBP-5** (Table 3), most likely due to intrinsic limitations of the MIC assay to discern small differences in activity. In contrast, for **ΔCPC'** a significant 2–4-fold decrease against Gram-negative bacteria was observed, while against Gram-positives activity remained mostly intact. These results suggest that recognition of LPS through the CPC' motif plays a critical role in the selective activity against Gram-negatives. An LPS-binding assay further illustrated distinctions between CPC'-defective peptides and **HBP-5**. Thus, $EC_{50}$ for either **R10** or **R14** was two- to fourfold higher than for **HBP-5**, and for **ΔCPC'** the increase was sevenfold. In addition, the CPC-modified peptides showed a substantial reduction in heparin binding, as shown by shorter retention times relative to **HBP-5**, notably a 20% decrease in % B elution buffer for **ΔCPC'** (Table 3; Fig. EV4). Finally, we inspected by CD the secondary structure of **R10**, **R14** and **ΔCPC'** in both LPS and heparin environments (Fig. EV4). While retaining an α-helical structure similar to **HBP-5**, the three CPC'-modified peptides exhibited decreased molar ellipticity maxima and minima, consistent with the LPS and heparin-binding results. Taken together, these findings corroborate the essential role of the CPC' motif in heparin and LPS binding, and its significant correlation with antimicrobial activity, as previously reported for peptide YI12WF (Pulido et al, 2017).

## Optimization of HBP-5 peptide and in vivo studies

To further explore the therapeutic potential of **HBP-5**, its stability to protease degradation was analyzed. The peptide was incubated in human serum and the time course of its degradation was monitored by HPLC to determine half-life (Table 4), which turned out to be relatively short ($t_{1/2} \sim 50$ min), thus limiting its therapeutic potential (Fig. 4A). Interestingly, in the HPLC analysis a proteolysis byproduct of about twice longer half-life was detected. The fragment, identified by MS as **HBP-5** minus a C-terminal tripeptide (Arg-Ser-Val) and accordingly named **HBP-5 [1–20]**, was synthesized de novo and a $t_{1/2}$ of ~88.5 min was determined for it (Fig. 4A). Given the modest $t_{1/2}$s of both peptides, we decided to explore the enantiomer of **HBP-5** (**dHBP-5**), a strategy previously applied with success (Sandín et al, 2021). Indeed, the switch to the all-D peptide extended $t_{1/2}$ to a satisfactory >360 min (Figs. 4A and EV5). Antimicrobial, cytotoxicity, hemolysis and LPS/heparin-binding data for **HBP-5 [1–20]** and **dHBP-5** are given in Table 4 and show both peptides behaving comparably to the parental **HBP-5**. Similarly, CD structural analysis in SDS, LPS and heparin (Fig. 4B) indicated a strong α-helical conformation upon binding, like the parental peptide.

The antimicrobial efficacy of **HBP-5** and its derived **HBP-5 [1–20]** and **dHBP-5** peptides was tested in vivo in a murine model

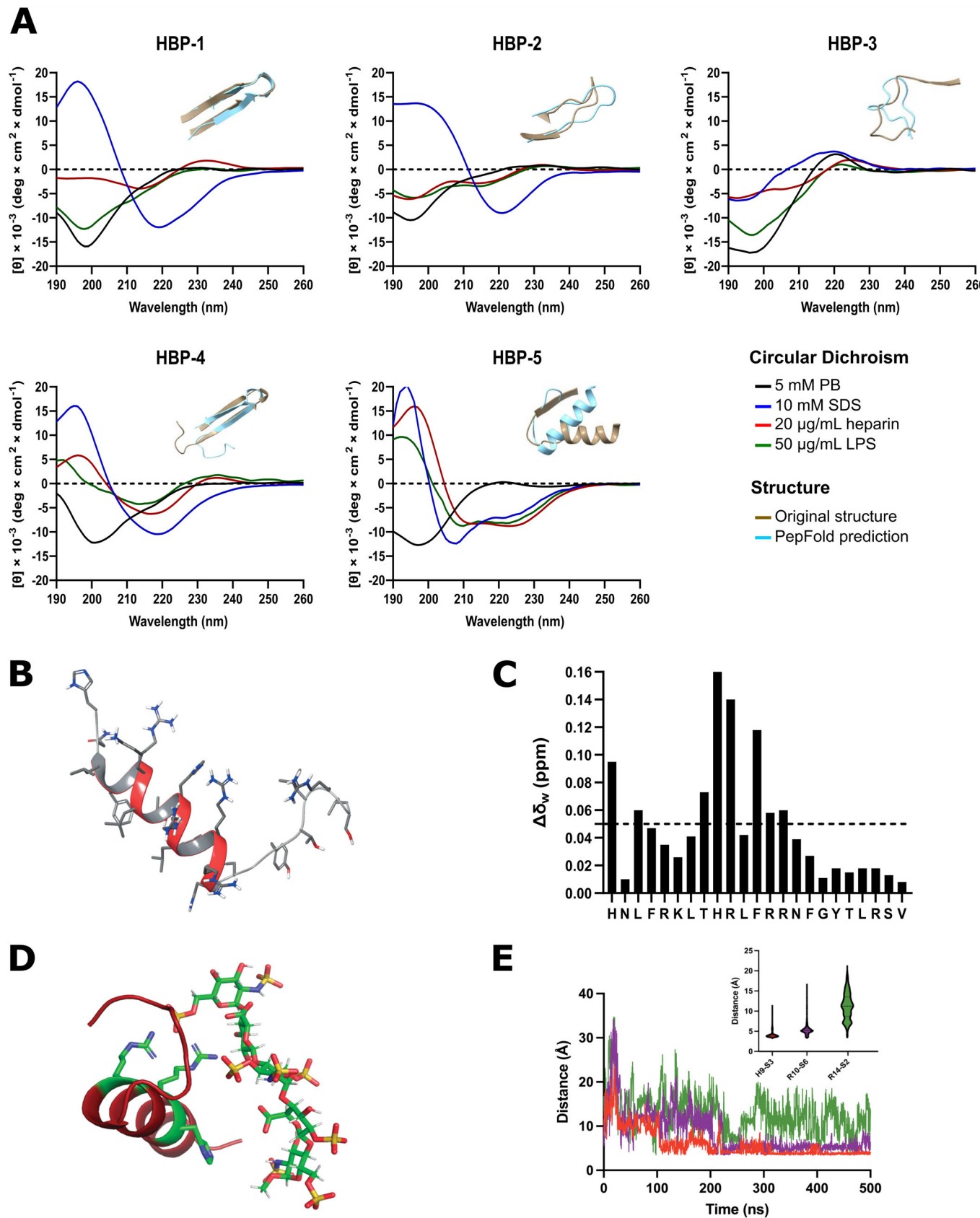

◀ **Figure 3. Structural characterization of HBP-5 in different conditions.**

(A) Circular dichroism spectra of HBPs in 5 mM PB (black lines), 10 mM SDS (blue lines), 20 µg/mL heparin (red lines), and 50 µg/mL LPS (green lines). Overlapping original peptide structures in native protein (in light blue) and PepFold predicted structures (in brown) are added for each peptide in the upper-right corner of each plot. (B) Structure of the peptide in DPC micelles as solved by NMR. (C) Weighted chemical shift differences ($\Delta\delta_w = [(\delta_{HN}^{bound} - \delta_{HN}^{free})^2 + (\delta_{Ha}^{bound} - \delta_{Ha}^{free})^2]^{1/2}$, ppm; see methods) induced by the presence of the heparin disaccharide H1S plotted as a function of peptide sequence. Peptide/disaccharide ratio 1:1. The horizontal line indicates the $\Delta\delta_w = 0.05$ ppm; Residues with values below this line are considered unaffected by interaction; and (D) structure of the peptide bound to the heparin analog Arixtra as defined by MD simulation. (E) Distances between the CPC' residues involved in heparin binding as observed in the MD simulation and summary of calculated distances. Source data are available online for this figure.

**Table 3. MIC and binding affinity of CPC' mutant peptides.**

| | | Peptide | | | |
|---|---|---|---|---|---|
| | | **HBP-5** | **R10** | **R14** | **ΔCPC'** |
| MIC (µM) | *E. coli* | 0.4 | 0.4 | 0.4 | 1.6 |
| | *A. baumannii* | 0.2 | 0.4 | 0.4 | 1.6 |
| | *Pseudomonas sp* | 0.8 | 0.8 | 0.4 | 1.6 |
| | *S. aureus* | 6.3 | 6.3 | 6.3 | 6.3 |
| | *L. monocytogenes* | 1.6 | 1.6 | 3.1 | 3.1 |
| Binding affinity | LPS (EC$_{50}$, µM) | 0.9 ± 0.7 | 3.9 ± 1.1 | 2.3 ± 1.0 | 6.7 ± 0.8 |
| | Heparin (% elution buffer) | 98.0 | 88.6 | 89.7 | 80.9 |

**Table 4. Properties of HBP-5-derived peptides.**

| | | Peptide | | |
|---|---|---|---|---|
| | | **HBP-5** | **HBP-5 [1–20]** | **dHBP-5** |
| MIC (µM) | *E. coli* | 0.4 | 0.4 | 0.1 |
| | *A. baumannii* | 0.2 | 0.4 | 0.2 |
| | *Pseudomonas sp* | 0.8 | 0.4 | 0.2 |
| | *S. aureus* | 6.3 | 6.3 | 0.8 |
| | *L. monocytogenes* | 1.6 | 1.6 | 0.4 |
| Binding affinity | LPS (EC$_{50}$ µM) | 0.9 ± 0.7 | 0.9 ± 0.5 | 1.1 ± 0.8 |
| | Heparin (% elution buffer) | 98.0 | 97.6 | 97.2 |
| Toxicity | Hemolysis (250 µM peptide) | 23 ± 1 | 24 ± 2 | 29 ± 1 |
| | EC$_{50}$ (MRC-5 cells, µM) | 69 ± 2 | 63 ± 1 | 30 ± 1 |
| Serum stability | Half-life ($t_{1/2}$, min) | 48.9 | 88.5 | >360 |

of sepsis induced by *Acinetobacter baumannii*. BALB/c mice were intraperitoneally inoculated with the bacterium, followed by a two-hour incubation to establish infection (Fig. 5A). Subsequently, three doses of peptide (or vehicle as control) were administered every 6 h. After 18 h, mice were euthanized, and colony-forming units (CFUs) in various organs were quantified to assess bacterial load. The results revealed a substantial CFU reduction (2–5 orders of magnitude) depending on organ and sex (Fig. 5B). Notably, the peptides outperformed positive control LL-37. Consistent with previous studies, female mice had a more efficient bacterial clearance than males (Carrera-Aubesart et al, 2024; Li et al, 2022). **HBP-5 [1–20]** showed particularly strong activity in females, with 4–8 order of magnitude CFU reduction across all tested organs

(Fig. 5B). This significant in vivo activity showcases **HBP-5 [1–20]** and **dHBP-5** as potential candidate AMPs for further preclinical development.

## Discussion

Inflammation and coagulation are closely related, with inflammatory proteins often interacting with GAGs and influencing anticoagulant activity (Sobczak et al, 2018). Some proteins play important roles in both processes, such as histidine-rich glycoprotein, an adapter protein released by platelets, that regulates angiogenesis, immunity, and coagulation (Sobczak et al, 2018).

Many proteins, particularly those involved in host defense, can act as reservoirs of AMPs, silently embedded in the protein sequences but produced on demand by host proteases during events such as inflammation, coagulation, etc (Autelitano et al, 2006; Kim et al, 2023; Mangoni et al, 2016; Torres et al, 2022). After a wound, processes to prevent bleeding, remove damaged tissue and keep the lesion free from pathogen entry and subsequent infection are called for (Thapa et al, 2020). In such scenarios, proteases hydrolyzing surrounding proteins and releasing (formerly) cryptic AMPs to achieve preventive antimicrobial action can play a crucial role (Sánchez et al, 2011). A relevant example is thrombin. While the whole protein does not display antimicrobial activity per se, after cleavage its C-terminus displays strong and broad activity (Papareddy et al, 2016). Computational tools have been recently developed to mine AMPs encrypted within proteins, some tools also predicting cleavage sites for proteolytic AMP release. The significant activity and therapeutic potential of encrypted AMPs has been demonstrated (Boaro et al, 2023; Santos et al, 2022; Torres et al, 2022).

It should therefore be not surprising that proteins involved in GAG binding can become an important source of AMPs, hence contribute to preventing infection (Ishihara et al, 2018; Malmström et al, 2009; Papareddy et al, 2016). This dual action, GAG binding and antimicrobial activity, can be interpreted in structural terms by the similarities between GAG and lipid A structures, both containing negatively charged disaccharide units. It is thus reasonable to propose that the ability to bind GAGs could also promote LPS recognition, hence allow interaction with the outer membrane of Gram-negative bacteria. The fact that LL-37 contains an XBBXBX-motif but lacks strong heparin-binding affinity (Table 1) and fails to show a significant preference for Gram-negative bacteria (Table 2), suggests that a CPC' motif may be relevant to bind both LPS and heparin, as shown by our structural analysis (Fig. 1). Heparin-binding and antimicrobial sequences are hence related, as suggested by similar amino acid compositions, but

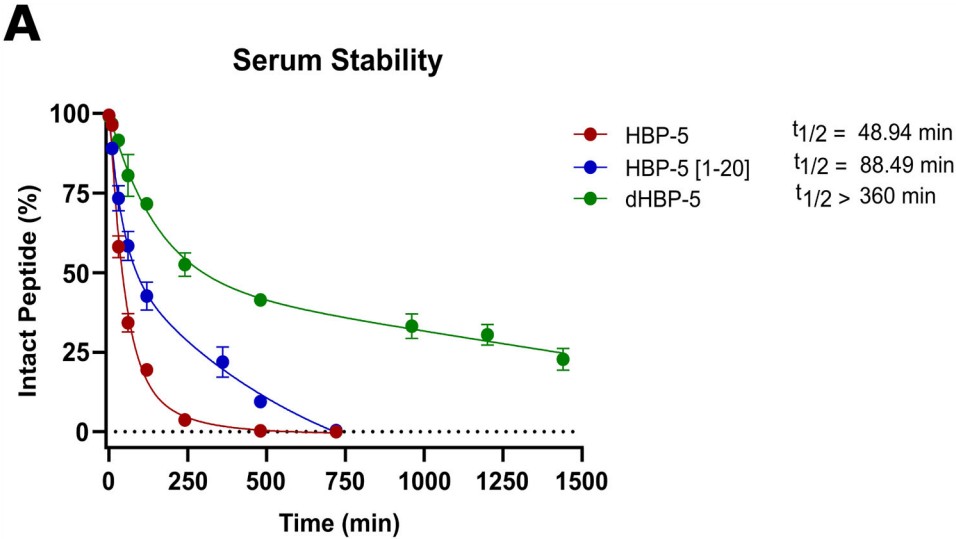

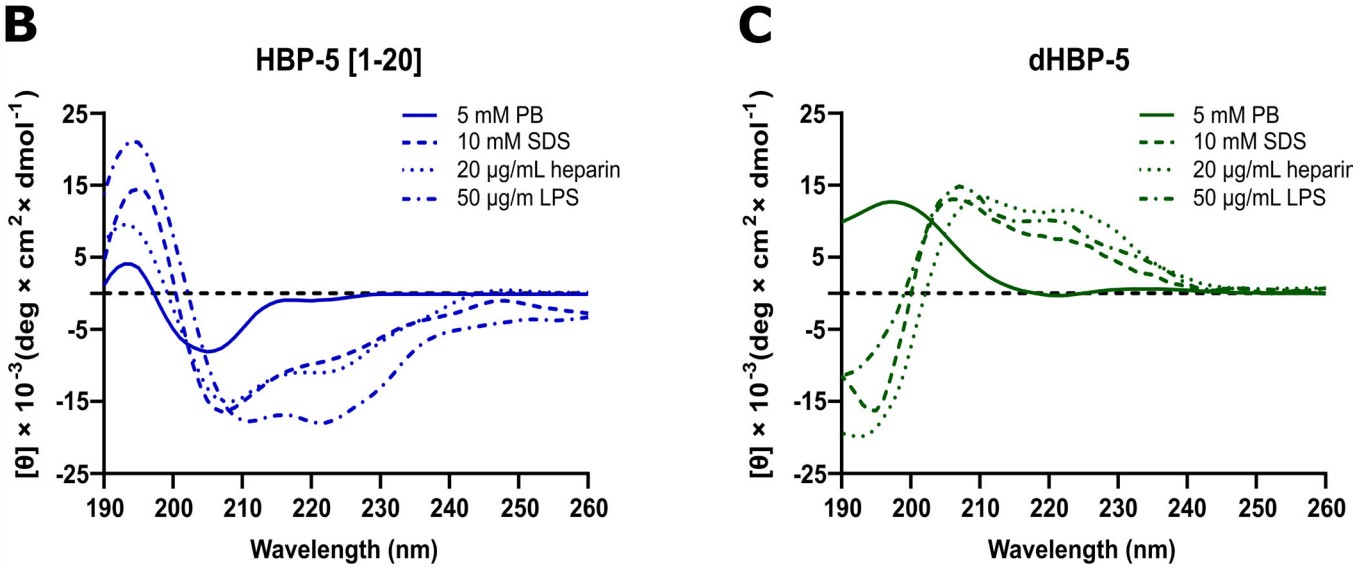

**Figure 4. Proteolytic stability and structural analysis of HBP-5 derived peptides.**

(**A**) Peptide extinction curves over time of peptides were obtained by integration of chromatogram peaks displayed in Fig. EV5. The peptide half-life ($t_{1/2}$) was estimated by experimental data fitting to an exponential decay model. Data are shown as the mean ± SEM of the three independent experiments. Circular dichroism spectra of HBP-5 [1-20] (**B**) and dHBP-5 (**C**) peptides in 5 mM PB, 10 mM SDS, 20 µg/mL heparin and 50 µg/mL LPS. Source data are available online for this figure.

a specific structural arrangement is clearly required to bind to LPS (Andersson et al, 2004; Kalle et al, 2014). This is further supported by the present results on **HBP-5** CPC' motif mutants, where deletion of the motif caused a significant drop in both LPS and heparin binding.

In conclusion, we have shown that GAG-binding proteins can be a source of new AMPs, some with remarkable activity. The fact that these peptides can bind both heparin and LPS is consistent with the above structural similarity hypothesis, and with the fact that these peptides have much higher activity on Gram-negative

bacteria. Results from in vivo data also show that AMPs derived from GAG-binding proteins, particularly peptide **HBP-5**, have strong activity in vivo and can reduce the bacterial load several orders of magnitude. Such results confirm that these peptides, if released upon injury due to protease cleavage could exert a significant bactericidal activity and contribute to innate immunity. Also, our results suggest that, with further optimization, HBP-encrypted AMPs should prove useful for treating infections by Gram-negative bacteria that are resistant to classic antibiotics and pose huge risks for hospitalized patients.

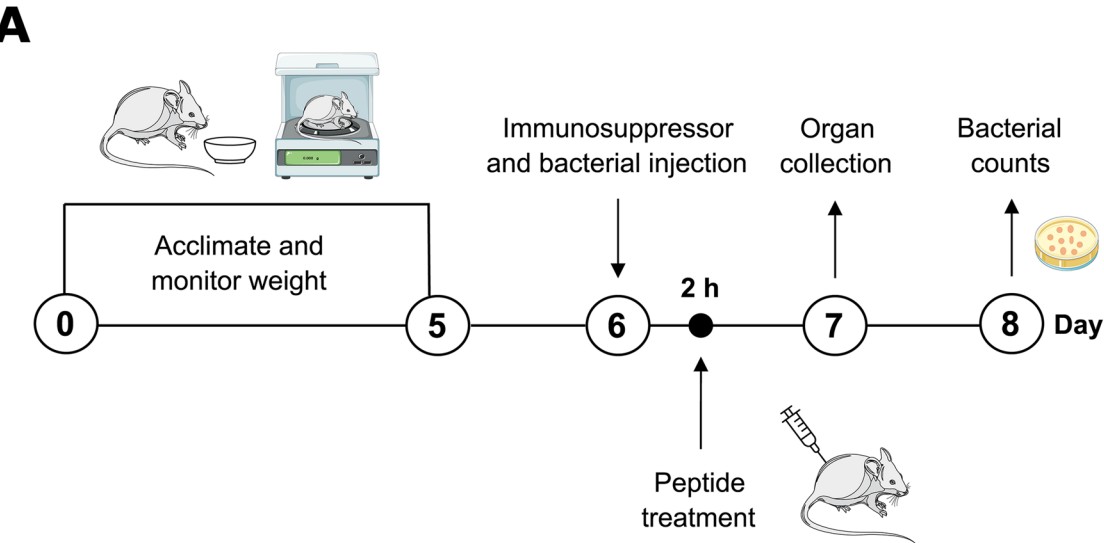

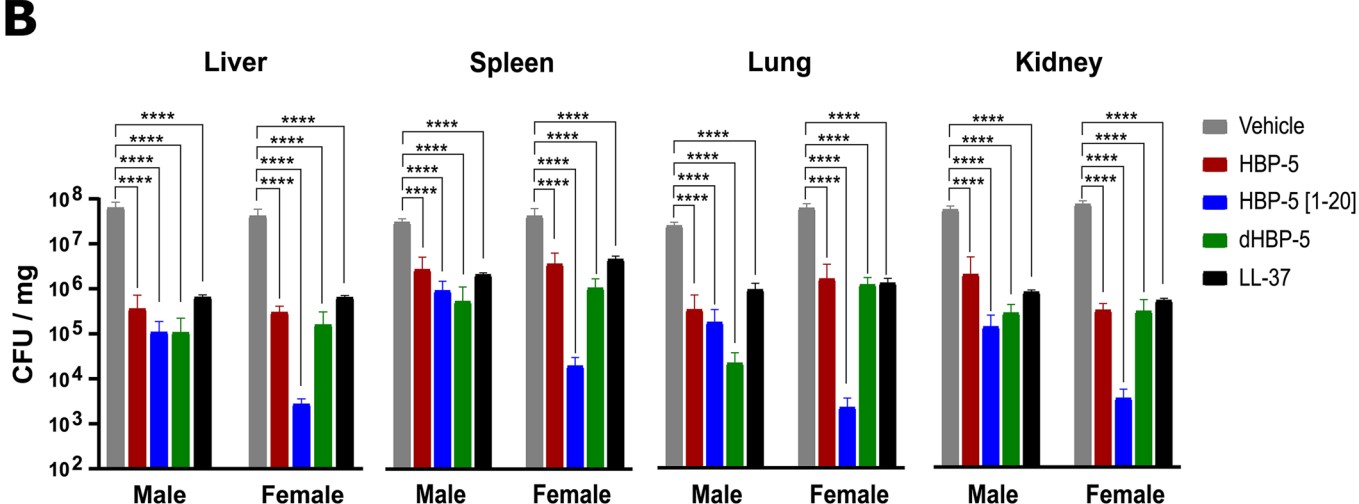

**Figure 5. Anti-infective activity of peptides in an animal model.**

(A) Schematic of the sepsis mouse model used to assess AMPs ($n = 6$) against *A. baumannii* ATCC 15308. (B) Average CFU/mg in organs after 18 h treatment. Statistical significance compared to vehicle was determined using two-way ANOVA followed by Dunnett's test (denoted as ****$P < 0.0001$). Exact *P* values are shown in Appendix Table S12. Source data are available online for this figure.

## Methods

**Reagents and tools table**

| Reagent/resource | Reference or source | Identifier or catalog number |
|---|---|---|
| **Experimental models** | | |
| *Escherichia coli* | CGSC | BW25113 7636 |
| *Acinetobacter baumannii* | CECT | ATCC 15308 |
| *Pseudomonas aeruginosa* | CECT | ATCC 15915 |
| *Staphylococcus aureus* | CECT | ATCC 12600 |
| *Enterococcus faecium* | CECT | ATCC 19434 |

| Reagent/resource | Reference or source | Identifier or catalog number |
|---|---|---|
| *Listeria monocytogenes* | CECT | ATCC 19112 |
| *Streptococcus pyogenes* | CECT | ATCC 8668 |
| HepG2 (Homo sapiens) | ATCC | HB-8065 |
| MRC-5 (Homo sapiens) | ATCC | CCL-171 |
| BALB/c Mice | Charles River Laboratory | 028 |
| Clinical strains | Vall d'Hebron Hospital | N/A |
| **Oligonucleotides and other sequence-based reagents** | | |
| Peptides | Synthesized | N/A |

| Reagent/resource | Reference or source | Identifier or catalog number |
|---|---|---|
| **Chemicals, enzymes, and other reagents** | | |
| Porcine mucin | Sigma | M2378 |
| Bovine Serum Albumin | Sigma | A7906 |
| NPN | Sigma | 104043 |
| TritonX-100 | Sigma | T8787 |
| Human Serum | Sigma | H4522 |
| *E. coli* LPS | Sigma | L2880 |
| Horse defibrinated red blood | Thermo Fisher | 10200013 |
| DiSC3(5) | Thermo Fisher | 18093953 |
| BODIPY-cadaverine | Thermo Fisher | 11500346 |
| MTT | Thermo Fisher | 11312727 |
| Trifluoroacetic acid | Thermo Fisher | 10294110 |
| Triisopropylsilane | Thermo Fisher | 10134650 |
| Acetonitrile | Thermo Fisher | 10660131 |
| HBS 1X | Thermo Fisher | 15313631 |
| HEPES | Thermo Fisher | 11422497 |
| MEMα | Thermo Fisher | 11524456 |
| Fetal bovine serum (FBS) | Thermo Fisher | 16629525 |
| Mueller-Hinton Broth | Thermo Fisher | 10681675 |
| Heparin disaccharide | TLC Pharmaceutical Standards | H-026006 |
| [D38]-DPC (98%) | Eurisotop | DLM-2274-0.1MG |
| $D_2O$ (99.9%) | Eurisotop | DLM-7005 |
| 96-well polypropylene plates | Greiner | 655261 |
| Luria Bertani Agar | Sudelab | 3161552 |
| **Software** | | |
| AutoDock Vina | https://vina.scripps.edu/ | |
| GraphPad Prism 8.0 | https://www.graphpad.com/ | |
| TOPSPIN software | https://www.bruker.com/ | |
| NMRFAM-SPARKY software | https://nmrbox.nmrhub.org/ | |
| OriginPro 2018 | https://www.originlab.com/ | |
| PyMOL | https://www.pymol.org/ | |
| GROMACS v2022.3 | https://www.gromacs.org/ | |
| **Other** | | |
| Prelude instrument | Gyros Protein Technologies | |
| Luna C18 column | Phenomenex | |
| Aeris Widepore XB-C18 column | Phenomenex | |
| LC-8 preparative RP-HPLC | Shimadzu | |

| Reagent/resource | Reference or source | Identifier or catalog number |
|---|---|---|
| LC-MS 2010EV | Shimadzu | |
| ÄKTA GO FPLC instrument | Cytiva | |
| Heparin HP column | Cytiva | |
| Tecan Spark instrument | Tecan | |
| Jasco J-815 spectropolarimeter | Jasco | |
| Varian Cary Eclipse spectrometer | Agilent | |
| EVO MA Electron Microscope | Zeiss Microscopy | |

## Antimicrobial activity prediction and docking

The library of heparin-binding proteins was obtained from previous studies (Ori et al, 2011). All sequences were processed with the AMPA antimicrobial peptide predictor (Torrent et al, 2012a) (http://tcoffee.crg.cat/apps/ampa, default parameters) to define antimicrobial regions. Best candidates by AMPA score were used for docking the heparin disaccharide H1S (α-ΔUA-2S-[1 → 4]-GlcNS-6S) in AutoDock Vina using default parameters (Eberhardt et al, 2021). Grid boxes were adjusted to the regions delineated by AMPA. To check for significant binding energy values, we docked H1S to the binding regions of proteins with a solved crystal structure containing a heparin analog. The proteins used as positive controls were angiogenin (4QFJ), heparin lyase I (3IN9), palmitoleoyl-protein carboxylesterase (4UYW), stromal cell-derived factor 1 (2NWG), peptidoglycan recognition protein 1 (3OGX), C–C motif chemokine 5 (1UL4), heparin cofactor 2 (1JMJ), antithrombin III (1SR5), annexin A2 (2HYV), plasma serine protease inhibitor (3DY0), and heparin lyase (2FUT). Crystal structures of non-heparin-binding proteins bound to a non-sulfated disaccharide were used as negative controls, i.e., aconitase (7ACN), R-methyltransferase (R30Q), phytase (3ZHC), bifunctional epoxide hydrolase 2 (1EK2), and calpain-3 (6BGP). The proteins included in the HBP list with the highest affinity score (above the average of positive controls) were checked for the presence of CPC' motifs within their sequence following previously reported criteria (Pulido et al, 2017; Torrent et al, 2012b).

## Peptides

Peptides were synthesized as described (Falcao et al, 2015) on H-Rink Amide-ChemMatrix resin in a Prelude instrument (Gyros Protein Technologies, Tucson, AZ) running Fmoc solid-phase peptide synthesis (SPPS) protocols. After sequence assembly, peptides were deprotected and cleaved in TFA/$H_2O$/triisopropylsilane (95:2.5:2.5 v/v), isolated by cold diethyl ether precipitation and centrifugation at 4800 rpm for 5 min, and lyophilized. Purification was performed on a Luna C18 column (21.2 mm × 250 mm, 10 μm; Phenomenex) in a LC-8 preparative RP-HPLC instrument (Shimadzu, Kyoto, Japan) using a linear gradient of solvent B (0.1% TFA in ACN) into A (0.1% TFA in $H_2O$) for 30 min at 25 mL/min flow rate. Prior and after purification, peptides were

inspected by analytical RP-HPLC and LC-MS. RP-HPLC was performed on a Luna C18 column (4.6 mm×50 mm, 3 μm; Phenomenex) in an LC-20AD instrument (Shimadzu) using a linear gradient of solvent B (0.036% TFA in ACN) into A (0.045% TFA in $H_2O$) over 15 min at 1 mL/min flow rate. LC-MS was done in an LC-MS 2010EV instrument (Shimadzu) fitted with an Aeris Widepore XB-C18 column (4.6 mm × 150 mm, 3.6 μm; Phenomenex) eluted with a liner gradient of solvent B (0.08% formic acid (FA) in ACN) into solvent A (0.1% FA in $H_2O$) over 15 min at 1 mL/min flow rate. Peptides with the expected mass and >95% HPLC homogeneity were lyophilized and stored at −20 °C. RP-HPLC chromatograms and ESI-MS spectra of peptides are shown in Appendix Figs. S4–S7.

## Heparin-binding affinity assay

Heparin binding was evaluated by affinity chromatography on a Heparin HP column (0.7 × 2.5 cm, 1 mL; Cytiva, Marlborough, MA) linked to an ÄKTA GO FPLC instrument (Cytiva, Marlborough, MA). In total, 10 mL of peptide stocks at 10 μM were loaded in the column, previously equilibrated with binding buffer (10 mM sodium phosphate). Peptides were eluted by a linear gradient of elution buffer (10 mM sodium phosphate, 2 M NaCl) into binding buffer. Heparin affinity for each peptide was defined as the percentage of elution buffer at maximum peak intensity.

## Minimum inhibitory concentration (MIC) and minimum bactericidal concentration (MBC)

Antimicrobial activities were determined by the microtiter broth dilution method recommended by the National Committee of Laboratory Safety and Standards (NCLSS), adapted for AMPs (Wiegand et al, 2008). Briefly, overnight bacterial cell cultures were brought to an exponential growth state ($OD_{600} = 0.4$) in MH broth and diluted to a final concentration of $5 \times 10^5$ CFU/mL. In all, 1:2 peptide serial dilutions were prepared in 96-well polypropylene plates (Greiner, Frickenhausen, Germany), in MH medium containing 0.2% (w/v) of bovine serum albumin (BSA) and 0.02% glacial acetic acid. Samples were incubated overnight at 37 °C and 230 rpm, and MIC was determined as the last peptide concentration without appreciable visual growth. MBC was determined by transferring 100 μL of the wells to Petri plates with Luria Bertani (LB) agar and incubated overnight at 37 °C. The lowest concentration with no colonies was taken as MBC. The results are the average of three independent studies.

## Killing curve assay

Antimicrobial activity was also tested by lethality curve in *E. coli* cultures (Sandín et al, 2022). First, 50 μL of peptide stock solution was added to 450 μL of an *E. coli* culture at $5 \times 10^5$ CFU/mL in a 1.5 mL polypropylene tube to obtain a final concentration of 1× MIC. Samples were then incubated at 37 °C and 600 rpm in an Accutherm microtube shaking incubator (Labnet, Edison, NJ) for 2 h. Samples of 50 μL were taken at several intervals and plated in LB agar. Plates were incubated overnight at 37 °C, and colonies were counted and compared with the initial inoculum to define the percentage of surviving bacteria. The results are the average of three independent studies.

## Hemolytic activity

Peptide toxicity was tested on horse erythrocytes (Sandín et al, 2021). Horse defibrinated blood was washed three times in phosphate buffer saline (PBS) pH 7.2 by centrifugation at 3× 3000 rpm for 10 min, to remove excess hemoglobin in the supernatant and then diluted tenfold in PBS. Next, 50 μL of erythrocytes were added to a 1.5 mL polypropylene tube and incubated with 50 μL of a 1:2 peptide serial dilution in PBS. An erythrocyte disruption (ED) control was prepared by adding 50 μL of 0.1% TritonX-100 in PBS instead of the peptides and an intact erythrocyte (IC) control by adding 50 μL of PBS alone. All samples and controls were incubated for 2 h at 37 °C. Samples were centrifuged at 3000 rpm for 3 min, and the supernatants were transferred to a polystyrene 96-well plate and inspected for ED by reading the absorbance at 540 nm in a Tecan Spark instrument (Tecan, Männedorf, Switzerland). The results are the average of three independent studies. The hemolysis percentage was calculated as:

$$Hemolysis(\%) = \frac{Sample - IC}{ED - IC} \times 100$$

## LPS-binding affinity

Displacement of fluorescent cadaverine bound to LPS was used to test peptide affinity (Torrent et al, 2011). Briefly, 50 μL of 1:2 peptide serial dilutions in 10 mM HEPES were prepared in polypropylene 96-well plates. Then, a previously incubated mixture of 25 μL of 40 μg/mL LPS and 25 μL of 40 μM cadaverine was added to each well. A control without peptides (NP) was prepared by the addition of 50 μL of HEPES buffer, and one without LPS (NL) by adding 25 μL of HEPES instead of LPS. Plates were read for fluorescence in a Tecan Spark instrument (Tecan, Männedorf, Switzerland), with excitation at 580 nm (5 nm slit) and emission at 620 nm (10 nm slit). The results are the average of three independent studies. The fraction of peptide bound to LPS was calculated as:

$$Binding = \frac{Sample - NP}{NL - NP}$$

## Bacterial membrane depolarization

DiSC3(5) lipophilic dye fixation was tested in *E. coli* fresh cultures to measure depolarization, as previously described (Torrent et al, 2009). In short, 5 mL bacterial suspensions in exponential phase (~0.4 OD) were washed first with 5 mL of buffer A (5 mM HEPES, 20 mM glucose, pH 7.2) and later with 5 mL of buffer B (5 mM HEPES, 20 mM glucose, 100 mM KCl, pH 7.2). Bacteria were then resuspended in buffer B to an OD of 0.05. In all, 1 mL samples were prepared and DiSC3(5) was added to a final concentration of 0.4 μM. Fluorescence emission was continuously measured in a Varian Cary Eclipse spectrometer (Agilent, Santa Clara, California), with excitation at 625 nm (5 nm slit) and emission at 666 nm (10 nm slit). 10 min after dye addition (estimated time for DiSC3(5) quenching), peptides were added to the samples to a final 10 μM concentration (except HBP-2, tested at 20 μM). Dye release was monitored for at least 5 min. The results are the average of three independent studies.

## Outer membrane permeabilization

Membrane permeability of peptides was measured by the uptake of 1-N-phenylnaphthylamine (NPN), as described (Ma et al, 2019). Briefly, bacterial suspensions of *E. coli* BW25113 were diluted to $10^8$ CFU/mL. The peptides were then added to a final concentration of 10 μM (except HBP-2, tested at 20 μM). Nontreated cells (addition of an equal volume of PBS) were used as negative control and LL-37 as positive control. Bacterial suspensions were incubated at 37 °C and 250 rpm, and 1 mL aliquots collected at different time points. Cells were washed twice and resuspended in 5 mM HEPES and 5 mM glucose buffer, pH 7.2, then incubated with NPN (10 μL from a 500 μM stock in acetone) for 15 min at 25 °C. Then, 200 μL of the samples were transferred to a 96-well black optical-bottom microplate. Fluorescence was measured at room temperature in a Tecan Spark instrument (Tecan, Männedorf, Switzerland) with excitation at 350 nm (10 nm slit) and emission at 420 nm (10 nm slit). The results are the average of three independent studies.

## Scanning electron microscopy (SEM)

In total, 1 mL of *E. coli* bacterial suspensions in exponential growth (~0.4 OD) were treated with 10 mM peptides for 2 h. After treatment, treated cells were filtered through a 0.1 mm Nucleopore filter to attach bacteria and later fixed for 2 h at 4 °C in a buffer containing 2.5% glutaraldehyde in 100 mM Na-cacodylate, pH 7.4. Afterward, cells were coated by immersion in 1% osmium tetroxide in Na-cacodylate buffer for 30 min. Samples were rinsed in the same buffer and dehydrated in ethanol with increasing concentrations (30 and 70% (v/v) once, 90 and 100% twice) for 15 min each. The filters were mounted on aluminum stubs and coated with gold-palladium in a sputter coater (K550; Emitech, East Grinsted, UK). Each sample was later inspected at 15 kV accelerating voltage in an EVO MA 10 scanning electron microscope (Zeiss, Oberkochen, Germany).

## Cytotoxicity in mammalian cells

Toxicity in MRC-5 and HepG2 cells was tested by the MTT assay, as described (Bello-Madruga et al, 2023). Cell lines were maintained in Eagle's minimum essential medium (MEMα) supplemented with 10% fetal bovine serum (FBS). Cells were cultured in 75-cm² flasks and then transferred to polystyrene 96-well plates, at $3 \times 10^4$ cells per well, and incubated overnight for attachment to the well surface. Then culture media was removed and 1:2 peptide serial dilutions in MEMα were added to each well and later incubated for 4 h. After incubation, peptides were removed and 100 μL of 0.5 mg/mL MTT staining solution in MEMα supplemented with 10% FBS was added to cells and incubated for 1.5 h at 37 °C. Formazan crystals in living cells were detected after disruption with 200 μL dimethyl sulfoxide and then absorbance was measured at 570 nm in a Tecan Spark instrument (Tecan, Männedorf, Switzerland). The results are the average of three independent studies.

## Circular dichroism

CD spectra of peptides were recorded in four different conditions: 5 mM phosphate buffer (PB), 5 mM PB with 10 mM SDS micelles, 5 mM PB with 20 μg/mL heparin, and 5 mM PB with 50 μg/mL LPS micelles. Peptides were dissolved in each condition to a final 10 μM concentration. Samples were transferred to a 0.1 mm quartz cuvette (Hellma, Jena, Germany) and then analyzed in a Jasco J-815 CD spectropolarimeter (Jasco, Tokyo, Japan). For each sample, 15 accumulation spectra were acquired from 260 to 190 nm at a scan speed of 100 nm/min. Data were processed with the OriginPro 2018 analysis software and subsequently analyzed to predict secondary structure with the CDSSTR method available at the DichroWeb online server (Miles et al, 2022) (http://dichroweb.cryst.bbk.ac.uk/html/home.shtml). Molar mean ellipticity [θ] was calculated as:

$$[\theta]_{MRW} \left( \deg \times cm^2 \times dmol^{-1} \right) = \frac{\theta(mdeg) \times MRW}{L(mm) \times c(mg/mL)}$$

where $[\theta]_{MRW}$ is the mean molar ellipticity by residue, MRW is the molecular weight of peptides divided by the number of residues minus one, θ is the ellipticity, L is the optical path (1.0 mm) and, c is the concentration.

## NMR spectroscopy

NMR samples were prepared by dissolving lyophilized peptide HBP-5 at about 1 mM concentration in aqueous solution ($H_2O$/$D_2O$ 9:1 v/v), in DPC micelles (30 mM [D38]-DPC in $H_2O$/$D_2O$ 9:1 v/v) or in aqueous solution with the heparin analog Arixtra or the heparin disaccharide H1S (molar ratios 1:1, 1:0.5). pH was measured using a glass micro-electrode and adjusted to 4.4 by the addition of NaOD or DCl. Sodium 2,2-dimethyl-2-silapentane-5-sulfonate (DSS) at a 0.1–0.2 mM concentration was added as the internal reference for the $^1$H chemical shifts. A Bruker AVNEO-600 spectrometer (Bruker Biospin, Karlsruhe, Germany) equipped with a cryoprobe was used to record NMR spectra: 1D $^1$H, 2D $^1$H,$^1$H-DFQ-COSY (double-filtered-quantum phase-sensitive two-dimensional correlated spectroscopy), $^1$H,$^1$H-TOCSY (total correlated spectroscopy), $^1$H,$^1$H-NOESY (nuclear Overhauser enhancement spectroscopy), and $^1$H-$^{13}$C-HSQC (heteronuclear single quantum coherence) at $^{13}$C natural abundance. TOCSY and NOESY mixing times were 60 ms and 150 ms, respectively. Data were processed using the TOPSPIN software (Bruker Biospin, Karlsruhe, Germany). The NMRFAM-SPARKY software (Lee et al, 2014) was used to analyze the NMR spectra. $^1$H chemical shifts were assigned by analysis of the 2D homonuclear spectra using the well-established sequential assignment methodology (Wishart et al, 1995), and $^1$H-$^{13}$C-HSQC spectra were analyzed to assign the $^{13}$C chemical shifts. The assigned chemical shifts have been deposited at the BioMagResBank (http://www.bmrb.wisc.edu) with accession codes BMRB ID: 51732 (HBP-5 in aqueous solution), 51740 (HBP-5/H1S 1:1) and 51767 (HBP-5 in DPC micelles).

The conformational shifts ($\Delta\delta_{H\alpha}$ and $\Delta\delta_{C\alpha}$) were obtained as the differences between the observed chemical shifts and those in random coil (RC) peptides: $\Delta\delta_{H\alpha} = \delta_{H\alpha}{}^{observed} - \delta_{H\alpha}{}^{RC}$, ppm and $\Delta\delta_{C\alpha} = \delta_{C\alpha}{}^{observed} - \delta_{C\alpha}{}^{RC}$, ppm $\delta_{H\alpha}{}^{RC}$ and $\delta_{H\alpha}{}^{RC}$ were taken from Wishart et al (Wishart et al, 1995). Helix populations (Appendix Table S10) were estimated from the $^1$Hα and $^{13}$Cα chemical shifts as previously described (Chaves-Arquero et al, 2020; Sandín et al, 2021). A weighted value for the chemical shift changes ($\Delta\delta w$, ppm)

was defined as:

$$\Delta\delta_W = [(\delta_{HN}^{bound} - \delta_{HN}^{free})^2 + (\delta_{H\alpha}^{bound} - \delta_{H\alpha}^{free})^2]^{1/2}$$

Considering all **HBP-5** residues (23 in total), the mean $\Delta\delta_w$ is 0.05 ppm. Residues with $\Delta\delta_w > 0.05$ ppm can be considered as those mostly affected by interaction.

The structure of **HBP-5** in DPC micelles was calculated using the iterative procedure for automatic NOE assignment integrated in the CYANA 3.98 program (Güntert, 2004). This algorithm consists of seven cycles of combined automated NOE assignment and structure calculation, in which 100 conformers were computed per cycle. The experimental input data comprises the lists of assigned chemical shifts, and integrated NOE cross-peaks present in the 150 ms NOESY spectra, plus the $\phi$ and $\psi$ dihedral angle restraints. The NOE cross-peaks were integrated using the automatic integration subroutine of the NMRFAM-SPARKY software (Lee et al, 2014). The TALOSn webserver (Shen and Bax, 2013) was used to obtain the dihedral angle restraints from the $^1$H and $^{13}$C chemical shifts. The final structure is the ensemble of the 20 lowest target function conformers calculated in the last cycle. These ensembles were visualized and examined by the MOLMOL program (Koradi et al, 1996).

### Molecular dynamics simulations

MD simulations with or without Arixtra were conducted using GROMACS v2022.3. The Glycan Reader & Modeler from CHARMM-GUI was used to prepare the system, obtaining the topology and parameter files. The force field CHARMM36 was employed for the protein and Arixtra parameters. Initial structures were solvated in a rectangular box of TIP3P water with a minimum distance of 1.0 nm between protein and the faces of the box. $K^+$ and $Cl^-$ ions were added to neutralize the system at an ionic strength of 0.15 M. Electrostatic interactions were calculated using the particle mesh Ewald method under periodic boundary conditions. Structures were energy-minimized and equilibrated by molecular dynamics for 130 ps. Production simulations were run on a GPU (NVIDIA GeForce RTX 3080 Ti) and 16 CPUs (Intel® Xeon® Gold 6226 R CPU @ 2.90 GHz) for 500 ns with a time step of 2 fs. NPT conditions were stabilized at 306 K by a V-rescale thermostat (Bussi et al, 2007), and at 1 atm by a Parrinello–Rahman barostat (Parrinello and Rahman, 1981). Bonds were constrained using the LINCS algorithm. Representative structures for different analyses were extracted from trajectories with the GROMACS command "gmx cluster", using the gromos algorithm with a RMSD cutoff of 0.18 nm.

### Serum stability

500 µL of 1 mM peptide stock in MiliQ water were mixed with 500 µL of human serum (Sigma-Aldrich, St. Louis, MO, USA). The mixture was incubated at 37 °C with shaking, and at various time points aliquots were taken and proteolysis was stopped with cold acetonitrile (80% in MiliQ water). The samples were chilled at 4 °C for 10 min to precipitate serum proteins, and centrifuged at 13,000 rpm at 4 °C for 15 min. The supernatant was collected and analyzed by LC and LC-MS with a –60% linear gradient of solvent

B (0.036%TFA in ACN) into A (0.045% TFA in H$_2$O) over 15 min at 1 mL/min flow rate. Unaltered peptide was determined by peak integration, expressed as the percent of the amount at 0 min, and data were fitted to a one-phase exponential decay model using GraphPad Prism 8.01 to estimate half-life time ($t_{1/2}$). The results are the average of three independent experiments.

### Mouse systemic infection

To determine the maximum lethal dose and appropriate doses for the main study, a toxicity study was first conducted. Peptides **HBP-5**, **dHBP-5**, and **HBP-5 [1–20]** were administered intraperitoneally (i.p.) at initial doses of 30 mg/kg. If toxicity signs or death occurred, a lower dose was administered to another animal. Once the adequate dose had been determined (**HBP-5** and **HBP-5 [1–20]**, 20 mg/kg; **dHBP-5**, 10 mg/kg), three randomly distributed groups of three female mice were treated thrice daily for three consecutive days to ensure that the selected doses did not induce toxicity.

Following previous protocols, a systemic infection model was established in mice (Carrera-Aubesart et al, 2024; Li et al, 2022). Briefly, an aliquot of *Acinetobacter baumannii* (ATCC 15308) stock stored at −80 °C in 15% glycerol was cultured overnight in 5 mL of MHB at 37 °C and 250 rpm shaking. The culture was then adjusted to an exponential growth phase (OD$_{600}$ = 0.3–0.6) and washed twice with sterile HBS 1×, pH 7.4, by centrifugation at 6000 rpm for 5 min. The cells were diluted with an identical volume of 10% porcine mucin solution to achieve a final concentration of $1 \times 10^8$ CFU/mL. Mucin enhances *A. baumannii* infectivity and acts as an immune suppressor (Carrera-Aubesart et al, 2024). 30 mice (15 male and 15 female) randomly distributed into five groups were each inoculated with a 10 mL/kg dose. Then, each group was treated i.p. with **HBP-5** and **HBP-5 [1–20]** at 20 mg/kg, **dHBP-5** at 10 mg/kg, **LL-37** at 5 mg/kg as positive control (Zarei-Mehrvarz et al, 2023), and HBS 1× vehicle as negative control. Treatment with three doses applied blindly every 6 h was started 2 h after infection. On day 2, animals were euthanized with a 200 mg/kg i.p. dose of pentobarbital. For end-of-treatment CFU evaluation, organs of interest (liver, spleen, lung, and kidney) were extracted, weighed, and homogenized in 1 mL HBS. Six serial tenfold dilutions were prepared, and each dilution was seeded (in triplicate) on Petri plates with LB agar. The colonies were counted after 16 h incubation at 37 °C. Final CFU counts are expressed as CFU/mg relative to organ weight.

## Data availability

This study includes no data deposited in external repositories. The source data of this paper are collected in the following database record: biostudies:S-SCDT-10_1038-S44320-025-00120-6.

## Peer review information

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

## Acknowledgements

Financial support from the Spanish Ministry of Economy and Innovation (PDC2021-121544-I00 funded by MCIN/AEI/10.13039/501100011033 and European Union Next Generation EU/ PRTR to MT; project PID2020-114627RB-I00 funded by MCIN/AEI/10.13039/501100011033 to MT; PID2020-112821GB-I00 funded by MCIN/AEI/10.13039/501100011033 to MAJ and PID2020-113184RB-C22 to DA), the Departament de Recerca i Universitats de la Generalitat de Catalunya (2023PROD00021 to MT), and from La Caixa Banking Foundation (HR17-00409, to DA). This work has been co-financed by the Spanish Ministry of Science and Innovation with funds from the European Union Next Generation EU, from the Recovery, Transformation and Resilience Plan (PRTR-C17.I1), European Society of Clinical Microbiology and Infectious Diseases (ESCMID) ESCMID2022 and from the Autonomous Community of Catalonia within the framework of the Biotechnology Plan Applied to Health. Work at UPF was funded by the MCI María de Maeztu network of Units of Excellence. NMR experiments were performed in the "Manuel Rico" NMR laboratory (LMR) of the Spanish National Research Council (CSIC), a node of the Spanish Large-Scale National Facility (ICTS R-LRB). DS and RBM are recipients of pre-doctoral a FPI scholarship (PRE2018-083243 and PRE2021-097678, respectively) from the Spanish Ministry of Science and Innovation.

## Author contributions

**Roberto Bello-Madruga**: Data curation; Formal analysis; Investigation; Methodology. **Daniel Sandín**: Data curation; Formal analysis; Investigation; Methodology. **Javier Valle**: Investigation; Methodology. **Jordi Gómez**: Data curation; Formal analysis; Investigation; Methodology. **Laura Comas**: Data curation; Formal analysis; Investigation; Methodology. **María Nieves Larrosa**: Resources; Writing—review and editing. **Juan José González-López**: Resources; Writing—review and editing. **María Ángeles Jiménez**: Resources; Supervision; Funding acquisition; Validation; Writing—original draft; Project administration; Writing—review and editing. **David Andreu**: Conceptualization; Resources; Supervision; Funding acquisition; Validation; Writing—original draft; Project administration; Writing—review and editing. **Marc Torrent**: Conceptualization; Resources; Supervision; Funding acquisition; Validation; Writing—original draft; Project administration; Writing—review and editing.

Source data underlying figure panels in this paper may have individual authorship assigned. Where available, figure panel/source data authorship is listed in the following database record: biostudies:S-SCDT-10_1038-S44320-025-00120-6.

## Disclosure and competing interests statement

The authors declare no competing interests.

# Expanded View Figures

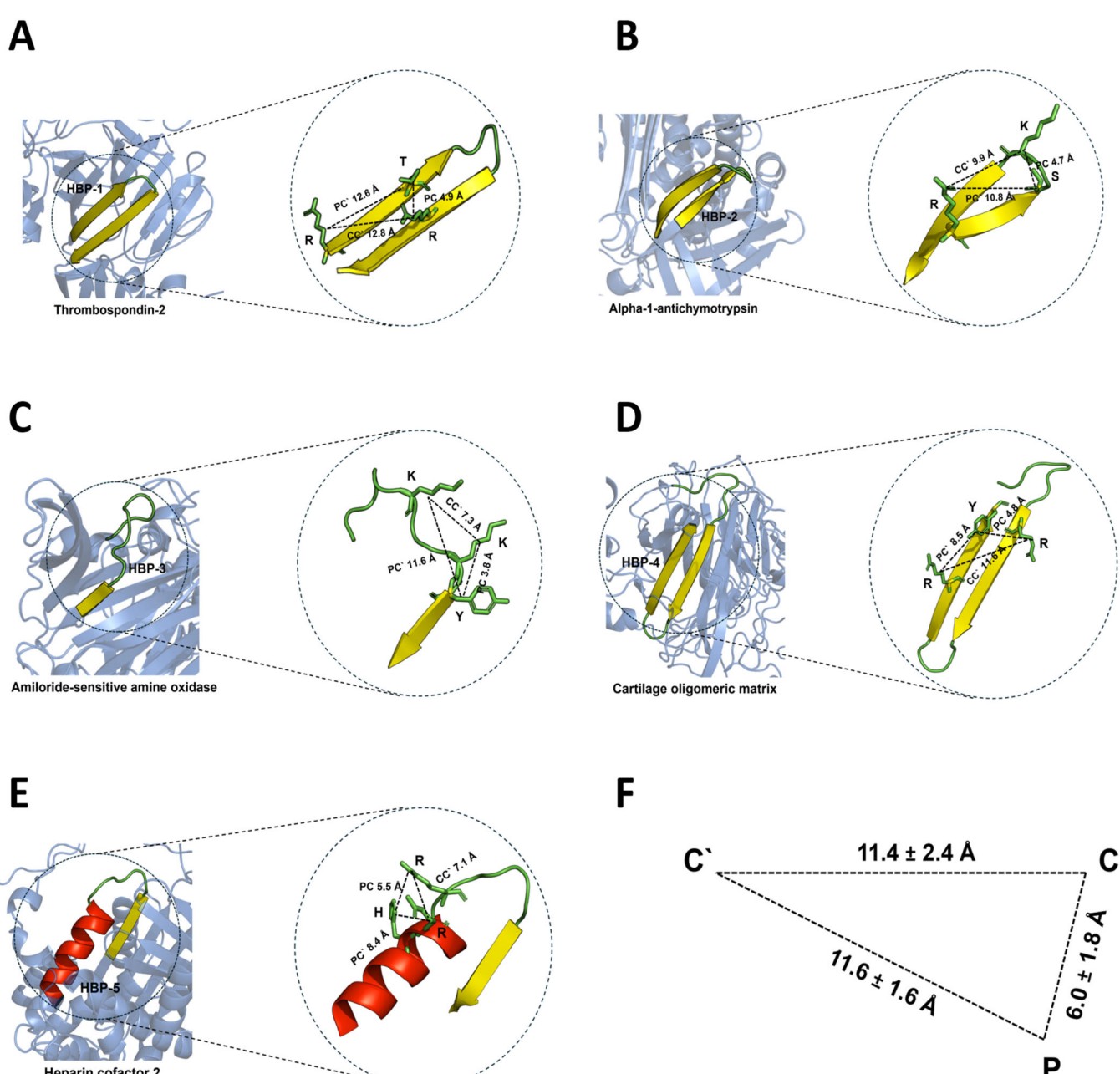

**Figure EV1. The CPC' clip motif of HBPs.**

Three-dimensional structure of the five selected HBPs and their respective CPC motifs in the context of the parent heparin-binding proteins: (A) HBP-1, (B) HBP-2, (C) HBP-3, (D) HBP-4, (E) HBP-5 and, (F) An outline of the CPC' clip motif. Tertiary structures are generated with Pymol. In the (A–E) CPC motifs helices are shown in red, β-strand in yellow, and loops are in green; remaining protein structures in transparent marine blue. The residues involved in the CPC' clip are connected with dashed lines.

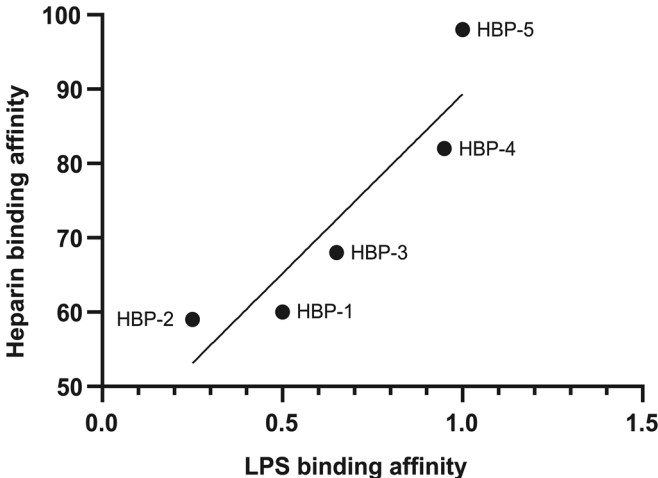

**Figure EV2.  Correlation between heparin and LPS-binding affinities of HBPs 1–5.**

Heparin affinity measured as the % of elution buffer required to dislodge the peptides from a heparin column; LPS affinity is measured as $EC_{50}$ values from the BODIPY-cadaverine assay. Source data are available online for this figure.

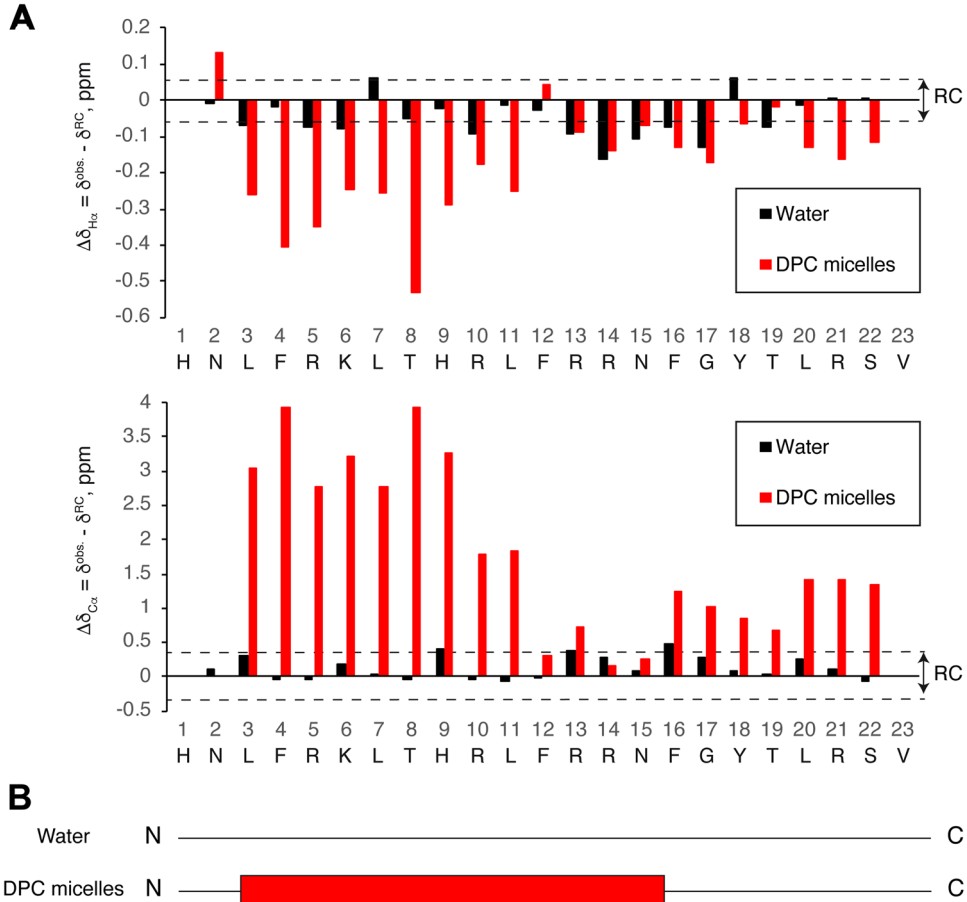

**Figure EV3. NMR chemical shifts.**

(A) $\Delta\delta_{H\alpha}$ and $\Delta\delta_{C\alpha}$ conformational shifts for HBP-5 in aqueous solution (black bars) and in DPC micelles (red bars) at pH 5.5 and 25 °C plotted as a function of peptide sequence. The two dashed lines indicate the random coil range (RC). (B) Schematic representation of the structural features in aqueous solution and in DPC micelles. Helices are shown as red rectangles. Source data are available online for this figure.

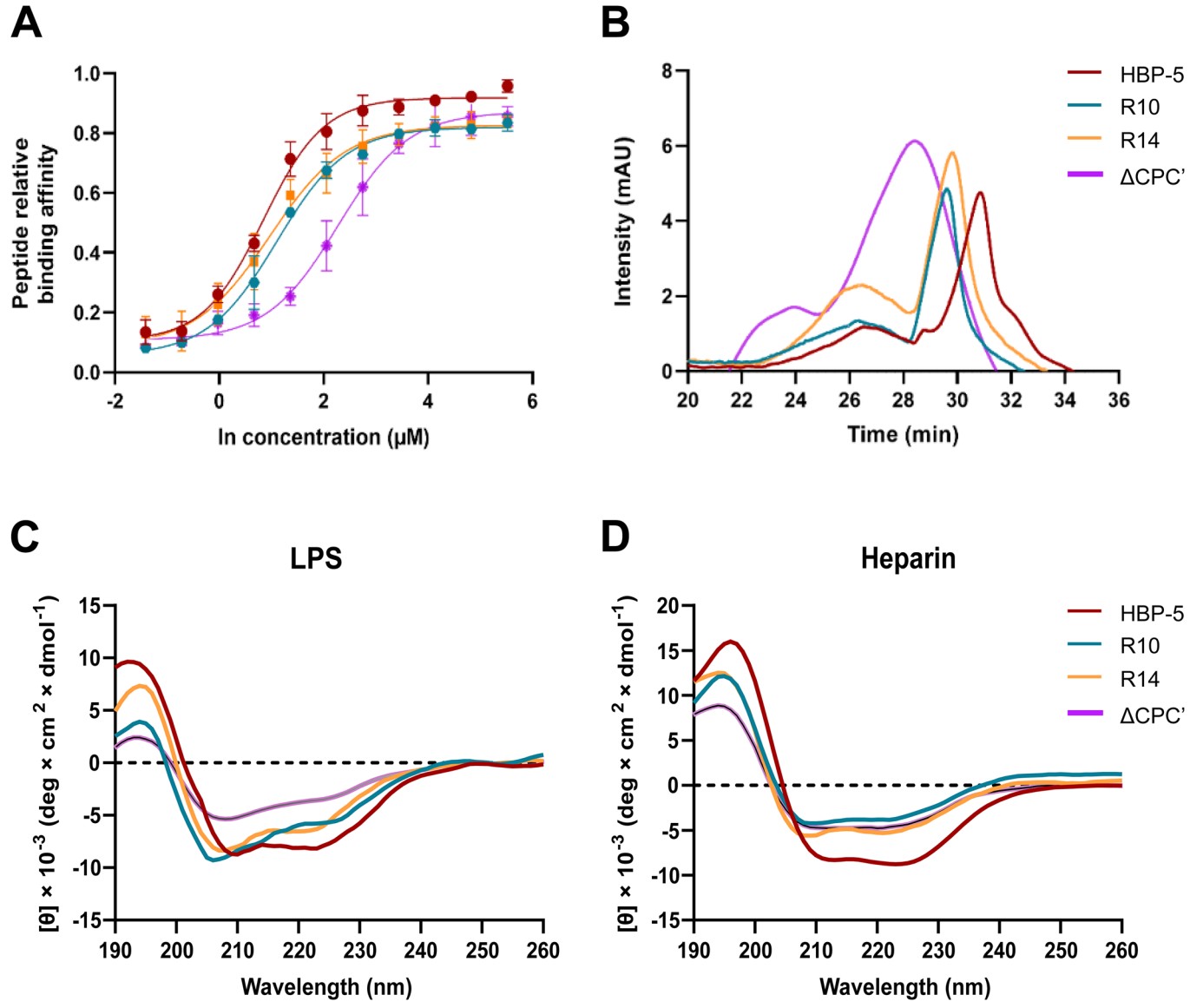

**Figure EV4. Structure-activity relationship study of HBP-5 and CPC' mutants.**

(A) LPS affinity measured as increase in fluorescence emission ($\lambda_{em} = 620$ nm) of BODIPY-cadaverine at different peptide concentrations. (B) Heparin-binding affinities in FPLC chromatography. (C, D) CD spectra of peptides in 50 µg/mL LPS (left) and 20 µg/mL heparin (right). Source data are available online for this figure.

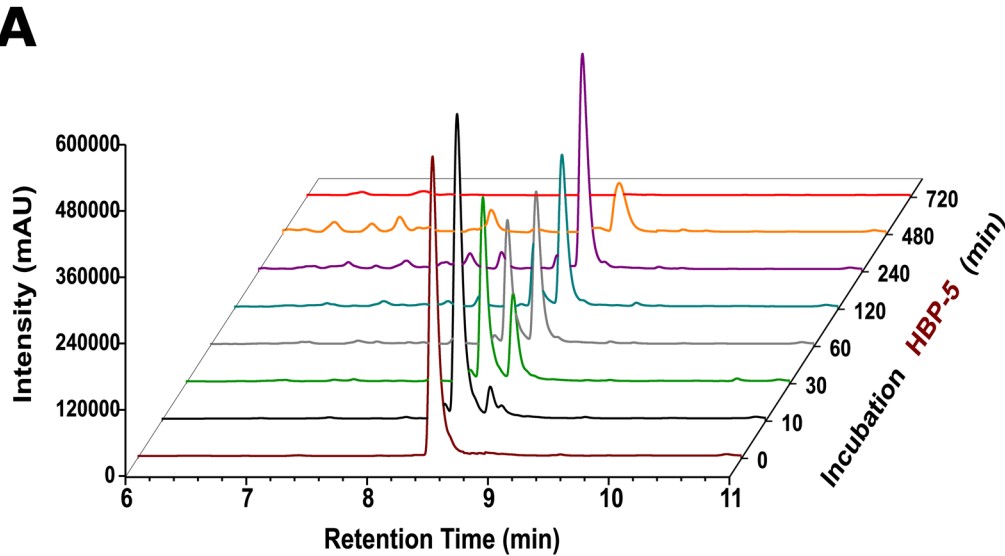

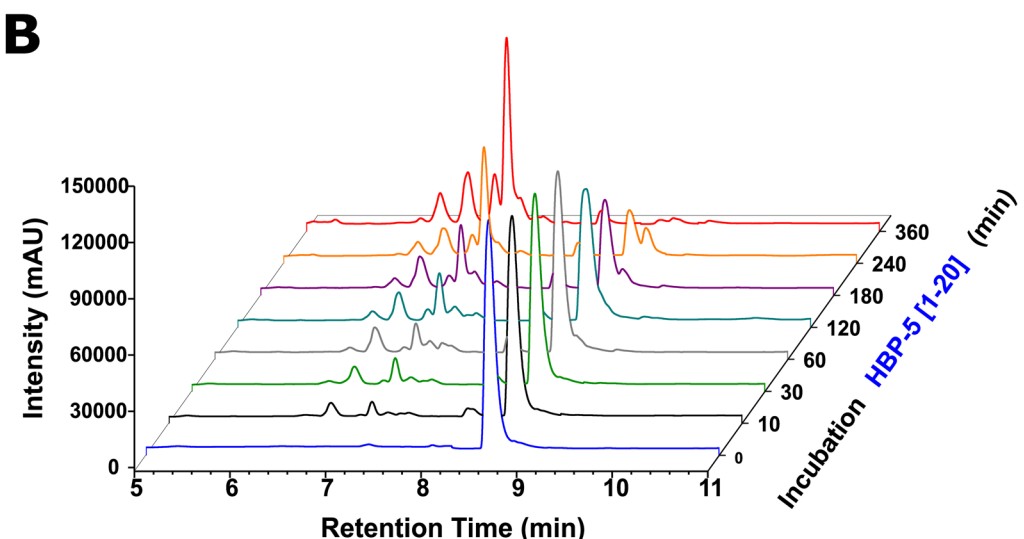

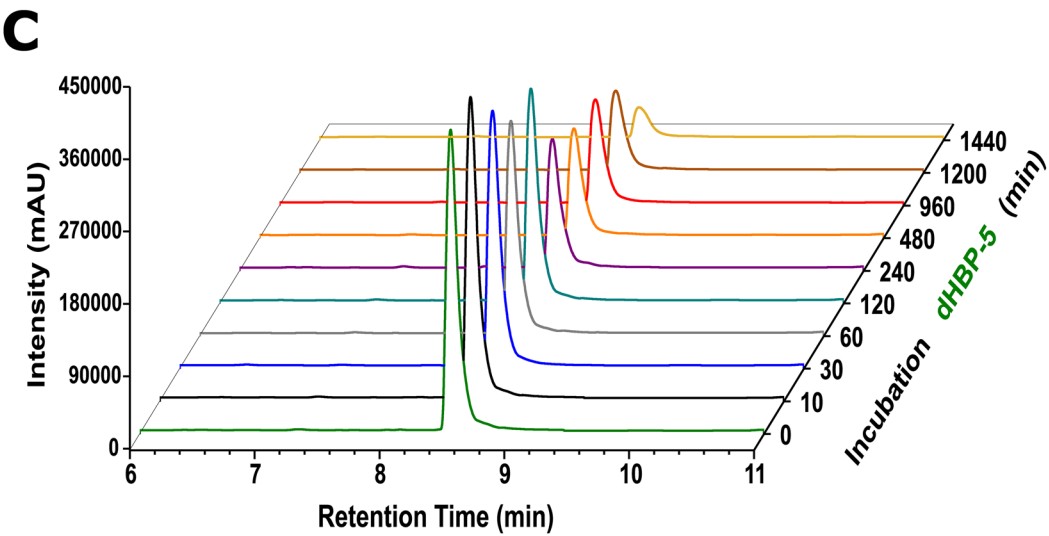

◀   **Figure EV5.   RP-HPLC chromatograms of peptides in human serum.**

Elution profiles after incubation of HBP-5 (**A**), HBP-5 [1–20] (**B**) and dHBP-5 (**C**) with 50% (v/v) human serum at representative times, using a 0% to 95% ACN gradient over 15 min. The peaks at $t_0$ correspond to the intact peptide. Source data are available online for this figure.

