## [Peer Review File · Molecular Systems Biology]

Mining the heparinome for cryptic antimicrobial peptides that selectively kill Gram-negative bacteria

Roberto Bello-Madruga, Daniel Sandín García, Javier Valle, Jordi Gomez Borrego, Laura Comas, Maria Nieves Larrosa, Juan José González, María Angeles Jiménez, David Andreu, and Marc Torrent

Corresponding author(s): Marc Torrent (marc.torrent@uab.cat) , David Andreu (david.andreu@upf.edu)

Review Timeline:

Transfer from Review Commons:	3rd Jun 24
Editorial Decision:	7th Jun 24
Appeal Received:	15th Jan 25
Editorial Decision:	4th Feb 25
Revision Received:	25th Mar 25
Accepted:	7th May 25

The logo for Review Commons, featuring the word "Review" in a large, blue, serif font with a diagonal slash through the letter 'v', and the word "COMMONS" in a smaller, blue, sans-serif font below it.

Editor: Jingyi Hou

Transaction Report: This manuscript was transferred to The EMBO JOURNAL following peer review at Review Commons.

Review #1

1. Evidence, reproducibility and clarity:

Evidence, reproducibility and clarity (Required)

Glycosaminoglycan (GAG)-binding proteins regulating essential processes such as cell growth and migration are essential for cell homeostasis. It is reported that the GAG has the ability to bind to Herpin sulfate. As both GAGs and the LPS lipid A disaccharide core of gram-negative bacteria contain negatively charged disaccharide units, the researchers proposed that heparin-binding peptides might have cryptic antimicrobial peptide motifs. To prove the hypothesis, they have synthesized five candidates [HBP1-5], which showed a binding affinity towards heparin and LPS binding. By using various methods, they showed that these molecules have antimicrobial activity. The key finding in this study is the finding of the CPC domain, where C is a cationic amino acid and P is a polar amino acid.

Major comments

1. Even though the Authors propose here that CPC' clip motif is needed for antimicrobial activity. However, various studies have demonstrated that the mere presence of cationic amino or hydrophobic amino acids does not give the activity, the location of these amino acids at the strategic position is critically needed. The major issue in this work, the authors have not presented, whether there was a single CPC motif or multiple in the 5 peptides they have synthesised. Further, they need to demonstrate how are the charged and hydrophobic amino acids distributed in the peptides. these things will clearly explain the difference in the activity as well spectrum of the peptides. The authors should make an extra figure or add information highlighting this unique characteristic for better understanding to the reader.
2. It is strange to observe that there are quite a number of reports showing that the peptides derived from the Herprin binding proteins have antimicrobial activity, but no one has reported their efficacy in the in vivo mouse model. if possible, the authors could add their observations if in vivo studies were done. or as a future line of study.

Minor comments

1. The presence of Cryptic antimicrobial domain in various heparin-binding proteins like laminin isoforms, von Willebrand factor, vitronectin, pro-tein C inhibitor, matrix glycoproteins thrombospondin, proline arginine-rich end leucine-rich repeat protein and fibronectin, have been reported previous. It is not clear why the authors did not refer to that

work. the authors should refer to the works.

****Referees cross-commenting****

****Minor comments****

1. The presence of Cryptic antimicrobial domain in various heparin-binding proteins like laminin isoforms, von Willebrand factor, vitronectin, pro-tein C inhibitor, matrix glycoproteins thrombospondin, proline arginine-rich end leucine-rich repeat protein and fibronectin, have been reported previous. It is not clear why the authors did not refer to that work. the authors should refer to the works. (same as reviewer 3)

All the earlier studies related to the antimicrobial activity of the peptides derived from the Heparin-binding protein reported a consensus Cardin and Weintraub motifs i.e, XBBBXXBX or XBBXBX, where X represents hydrophobic or uncharged amino acids, and B represents basic amino acids. However, in this work, the researchers report about the presence of the new CPC motif. So this is unique and a novelty in the study.

Even though the researchers report on the role of the CPC motif in the antimicrobial activity and binding to the heprin, the authors did not show any data or draw the conclusions related to the CPC domain when it comes to differences in the activity. this is the weakness of the manuscript. (same as reviwier 2)

2. It is strange to observe that there are quite a number of reports showing that the peptides derived from the Herprin binding proteins have antimicrobial activity, but no one has reported their efficacy in the in vivo mouse model. if possible, the authors could add their observations if in vivo studies were done. or as a future line of study.(Same as reviewer 2)

2. Significance:

Significance (Required)

All the earlier studies related to the antimicrobial activity of the peptides derived from the Heparin-binding protein reported a consensus Cardin and Weintraub motifs i.e, XBBBXXBX or XBBXBX, where X represents hydrophobic or uncharged amino acids, and B represents basic amino acids. However, in this work, the researchers report about the presence of the new CPC motif. So this is unique and a novelty in the study.

Even though the researchers report on the role of the CPC motif in the antimicrobial activity and binding to the heprin, the authors did not show any data or draw the conclusions related to the CPC domain when it comes to differences in the activity. this is the weakness of the manuscript.

There are more than 20 different AMP databases or prediction software. however, not all of them are 100 % current, their success rate varies from 30-50% only. It needs to be investigated if adding this search in the hit peptides might increase the success rate of the extra in silico-based AMPs prediction software

3. How much time do you estimate the authors will need to complete the suggested revisions:

Estimated time to Complete Revisions (Required)

(Decision Recommendation)

Less than 1 month

4. Review Commons values the work of reviewers and encourages them to get credit for their work. Select 'Yes' below to register your reviewing activity at Web of Science Reviewer Recognition Service (formerly Publons); note that the content of your review will not be visible on Web of Science.

Yes

Review #2

1. Evidence, reproducibility and clarity:

Evidence, reproducibility and clarity (Required)

This is a very nice paper by the Andreu and Torrent groups that report the antimicrobial and heparin-binding of several encrypted peptides. Overall, this study presents an intriguing exploration into the potential dual functionality of glycosaminoglycan (GAG)-binding proteins, specifically heparin-binding proteins (HBPs), in recognizing lipopolysaccharide (LPS) and exhibiting antimicrobial properties. The findings, particularly the identification and characterization of novel encrypted peptides, such as HBP-5, are promising and

contribute to our understanding of the intricate interplay between GAG-binding proteins and immunity. The data provided and methodology are thorough and well described. In sum, this is a very nice work. Please see below my minor comments.

****Minor comments:****

- Fig. 1 legend does not show antimicrobial activity. Please remove from the figure legend title.
- Discussion section: the authors should expand this section a bit to discuss recent work in the encrypted/cryptic peptide area. There are some recent relevant papers published in the past 3 years that should be discussed.
- References provided are a bit outdated and do not accurately reflect the latest in the field (see comment above).
- Gram should be capitalized throughout the text.
- Can the authors comment on the potential translatability of HBP-5? Please also comment on the potential advantages of having peptides that 1) bind to heparin; and 2) kill bacteria.
- More details on the computational tools and methods used to mine the peptides are needed.

2. Significance:

Significance (Required)

The data provided and methodology are thorough and well described. In sum, this is a very nice work.

3. How much time do you estimate the authors will need to complete the suggested revisions:

Estimated time to Complete Revisions (Required)

(Decision Recommendation)

Less than 1 month

Yes

Review #3

1. Evidence, reproducibility and clarity:

Evidence, reproducibility and clarity (Required)

****Summary:**** This manuscript has identified and investigated antimicrobial peptides from GAG binding proteins. Authors hypothesized that due to physiochemical similarity between GAG and LPS, fragments of GAG binding proteins might exert antimicrobial activity particularly against G- bacteria. Authors have identified few such AMPs that demonstrate LPS binding and displayed antibacterial activity. They have also solved NMR structure of the potent peptide and mode of action.

****Major comments:**** AMPs are promising molecules that can serve as lead for the development of therapeutics against MDR bacteria. In particular, currently therapeutic options to treat MDR Gram negative pathogens are limited. The current study is interesting and provides new non-toxic AMPs. Conclusions drawn from the works are largely valid. However, authors should address following comments

1. The design and characterization of the peptide YI12WF is not described. Previous studies had shown design of b-boomerang peptides (Bhattacharjya and coworkers) that target LPS.
2. Mutations or substitution of the key residues peptide 5 might improve the novelty of the work.
3. How these peptides disrupt LPS permeability is not investigated. As LPS is the major target.
4. Are the D-enantiomers of the peptides active against bacteria?
5. 3-D structure of peptide 5 is solved in DPC micelle which is a mimic for eukaryotic cells. Authors should attempt to determine structure in LPS as shown in several recent studies with potent AMPs thanatin, MSI etc,

****Minor comments:**** There are examples of AMPs derived from human proteins. Authors should highlight such works.

2. Significance:

Significance (Required)

The work described in the manuscript is novel and hold promises to develop antimicrobials in future.

3. How much time do you estimate the authors will need to complete the suggested revisions:

Estimated time to Complete Revisions (Required)

(Decision Recommendation)

Between 3 and 6 months

No

Response to Reviewer 1

Glycosaminoglycan (GAG)-binding proteins regulating essential processes such as cell growth and migration are essential for cell homeostasis. It is reported that the GAG has the ability to bind to Heparin sulfate. As both GAGs and the LPS lipid A disaccharide core of gram-negative bacteria contain negatively charged disaccharide units, the researchers proposed that heparin-binding peptides might have cryptic antimicrobial peptide motifs. To prove the hypothesis, they have synthesized five candidates [HBP1-5], which showed a binding affinity towards heparin and LPS binding. By using various methods, they showed that these molecules have antimicrobial activity. The key finding in this study is the finding of the CPC domain, where C is a cationic amino acid and P is a polar amino acid.

Major comments

1. Even though the Authors propose here that CPC' clip motif is needed for antimicrobial activity. However, various studies have demonstrated that the mere presence of cationic amino or hydrophobic amino acids does not give the activity, the location of these amino acids at the strategic position is critically needed. The major issue in this work, the authors have not presented, whether there was a single CPC motif or multiple in the 5 peptides they have synthesized. Further, they need to demonstrate how are the charged and hydrophobic amino acids distributed in the peptides. these things will clearly explain the difference in the activity as well spectrum of the peptides. The authors should make an extra figure or add information highlighting this unique characteristic for better understanding to the reader.

We thank the reviewer for his/her comments and suggestions. We concur that the distribution of amino acids is crucial for the antimicrobial activity of the peptides and their ability to bind heparin. We also agree with the suggestion of illustrating the location of the CPC' motifs of HBPs in the context of the parental proteins and have accordingly done so in the new Supplementary Figure 1. In all cases, only one CPC' motif was identified in the antimicrobial region, as highlighted in the figure, and the inter-residue distances measured are consistent with the CPC' motif definition. Thus, we demonstrate that a CPC' motif exists in all five HBPs, which explains how they recognize and bind heparin.

To illustrate the distribution of charged and hydrophobic amino acids in HBPs, we have also prepared new Supplementary Figure 2, displaying electrostatic potentials in the predicted HBP structures, and showing how the distribution of charged residues creates hydrophobic and cationic patches on the surface of the peptides. Our analysis reveals cationic patches to be surrounded by hydrophobic residues, which may explain the ability of the peptides to disrupt membranes and exert antimicrobial activity.

2. It is strange to observe that there are quite a number of reports showing that the peptides derived from the Heparin binding proteins have antimicrobial activity, but no one has reported their efficacy in the in vivo mouse model. if possible, the authors could add their observations if in vivo studies were done. or as a future line of study.

We thank the reviewer for his/her comment on the observation of antimicrobial activity in peptides derived from heparin-binding proteins. Indeed, a few such studies have appeared in the literature, with little success [1]. We believe this is due to a lack of understanding of how to identify heparin-binding regions in proteins and AMPs, which underlies their relative paucity. In this context, we anticipate that our results will spur further efforts and increase success, specifically by providing a rationale on how to identify CPC' motifs hence heparin-binding regions in protein sequences.

Regarding the suggestion of assessing the *in vivo* efficacy of HBPs, we would agree that it would be helpful for better understanding their potential therapeutic applications. However, we feel that such experiments are beyond the scope of our manuscript, which offers ample, compelling *in vitro* and *in silico* evidence of how heparin-binding proteins can be a source of AMPs. We have done this by showing that CPC' motifs embedded in such proteins can be unveiled, accurately defined in structural terms, and experimentally shown to possess antimicrobial activity. Furthermore, we have shown that heparin binding correlates with LPS binding, allowing us to propose a mechanistic explanation for how heparin binding can be related to antimicrobial activity.

Translating these results to animal models is possibly premature at this stage as, from a classical medicinal chemistry perspective, it would require previous structural elaboration in terms of, e.g., optimized serum half-life or serum protein binding, both of which can modulate activity in *in vivo* studies regardless of heparin affinity or bactericidal activity per se. Ongoing work in our laboratories is focused in these directions and will be reported in due time.

Referees cross-commenting*

Minor comments

1. The presence of Cryptic antimicrobial domain in various heparin-binding proteins like laminin isoforms, von Willebrand factor, vitronectin, protein C inhibitor, matrix glycoproteins thrombospondin, proline arginine-rich end leucine-rich repeat protein and fibronectin, have been reported previous. It is not clear why the authors did not refer to that work. The authors should refer to the works. (same as reviewer 3)

We were aware of other prior studies on heparin-binding proteins and did indeed cite some of them, though not exhaustively for conciseness' sake. However, as encouraged by reviewers 1 and 3 we have cited the following studies:

Malmström E, Mörgelin M, Malmsten M, Johansson L, Norrby-Teglund A, Shannon O, Schmidtchen A, Meijers JC, Herwald H. Protein C inhibitor--a novel antimicrobial agent. PLoS Pathog. 2009, 5:e1000698.

Ishihara, J., Ishihara, A., Fukunaga, K. et al. Laminin heparin-binding peptides bind to several growth factors and enhance diabetic wound healing. Nat Commun. 2018, 9:2163

Chillakuri Chandramouli R, Jones Céline and Mardon Helen J. Heparin binding domain in vitronectin is required for oligomerization and thus enhances integrin mediated cell adhesion and spreading, FEBS Letters. 2010, 584:3287-91.

Papareddy P, Kasetty G, Kalle M, Bhongir RK, Mörgelin M, Schmidtchen A, Malmsten M. NLF20: an antimicrobial peptide with therapeutic potential against invasive Pseudomonas aeruginosa infection. J Antimicrob Chemother. 2016, 71:170-80.

All the earlier studies related to the antimicrobial activity of the peptides derived from the Heparin-binding protein reported a consensus Cardin and Weintraub motifs i.e, XBBBXXBX or XBBXBX, where X represents hydrophobic or uncharged amino acids, and B represents basic amino acids. However, in this work, the researchers report about the presence of the new CPC motif. So, this is unique and a novelty in the study.

We thank the reviewers for these observations. Indeed, our quest to unveil CPC' motifs in antimicrobial regions of heparin-binding proteins is the key point of our investigation, and what distinguishes it from previous studies on consensus motifs such as XBBBXXBX or XBBXBX. We believe our definition of CPC' motifs in simple, structure-based, and experimentally verifiable terms is not only a significant departure but also a step forward from earlier views, highlighting the importance of a structural perspective in defining heparin-binding regions. In

point of fact, we show that our peptides, even without consensus Cardin-Weintraub motifs, bind heparin with high affinity. The presence of the CPC' motif is crucial for such binding, as well as for LPS binding, and the new experiments performed at editor/reviewer's request, where the CPC motif in HBP5 is abolished, with predictable impact, fully support our view, see new section "Insights into the CPC' motif of HBP-5 and its implication on the antibacterial mechanism" and new Table 3 in the revised manuscript.

2. Even though the researchers report on the role of the CPC motif in the antimicrobial activity and binding to the heparin, the authors did not show any data or draw the conclusions related to the CPC domain when it comes to differences in the activity. this is the weakness of the manuscript. (same as reviewer 2)

We welcome the reviewer's observation. To address it, we made and tested three HBP-5 mutants aimed at showing how alterations in the CPC' motif might influence interaction with heparin and LPS, as well as antimicrobial properties. The first two mutants involved replacing positively charged R10 and R14 residues with glutamine, similar in size and polarity but uncharged. As shown in the new section "Insights into the CPC' motif of HBP-5 and its implication on the antibacterial mechanism" and on the new Table 3 of the revised manuscript, the changes reduced heparin binding, i.e., shorter retention times on affinity chromatography, as well as LPS binding, i.e., a decrease in EC₅₀ in the cadaverine assay (Table 3). The modifications had a lesser impact on antimicrobial activity, most likely due to the low resolution of MIC assays.

In a further step to assess the effect of the CPC' motif on antimicrobial activity, we deleted it in full by replacing residues H9, R10 and R14 of HBP-5 by alanine. As expected, this Δ CPC' peptide showed a sharp reduction in both heparin and LPS binding (Table 3) and, most importantly, a significant and asymmetric change in antimicrobial activity, with substantial impact on Gram-negatives yet practically no effect on Gram-positives, suggesting that LPS plays a key role in this selective response. Altogether, these observations align with our hypothesis that heparin-binding proteins might exploit their intrinsic affinity for heparin as an opportunity to developing antimicrobial properties by leveraging structural similarities between glycosaminoglycans and LPS.

3. It is strange to observe that there are quite a number of reports showing that the peptides derived from the Heparin (*sic*) binding proteins have antimicrobial activity, but no one has reported their efficacy in the in vivo mouse model. if possible, the authors could add their observations if in vivo studies were done. or as a future line of study. (Same as reviewer 2)

We would kindly direct attention to #2 in the response to reviewer 1 above.

4. There are more than 20 different AMP databases or prediction software. however, not all of them are 100 % current, their success rate varies from 30-50% only. It needs to be investigated if adding this search in the hit peptides might increase the success rate of the extra in silico-based AMPs prediction software.

If we understand the question correctly, the reviewer wonders whether including a CPC' motif predictor would increase the accuracy of AMP search algorithms. In our view, this strategy has two main limitations to be considered: (i) locating a CPC' motif in a peptide sequence typically requires a known 3D structure. Unfortunately, this is not always the case, and for proteins lacking reliable 3D data it can be a challenging and resource-intensive process; (ii) while CPC' motifs may predispose proteins to evolve antimicrobial properties, it is unclear if this is a required feature for all AMPs. Imposing the presence of a CPC' motif may not be applicable to

all AMPs, although it might help identifying peptides with specific activity against gram-negative strains.

In summary, while the query of including a CPC' motif search tool in AMP predictors is intriguing and worthy of exploration for its potential bearing on antimicrobial research, it is technically complicated and beyond the scope of our manuscript.

Reviewer #1 (Significance (Required)):

All the earlier studies related to the antimicrobial activity of the peptides derived from the Heparin-binding protein reported a consensus Cardin and Weintraub motifs i.e, XBBBXXBX or XBBXBX, where X represents hydrophobic or uncharged amino acids, and B represents basic amino acids. However, in this work, the researchers report about the presence of the new CPC motif. So this is unique and a novelty in the study.

Even though the researchers report on the role of the CPC motif in the antimicrobial activity and binding to the heparin, the authors did not show any data or draw conclusions related to the CPC domain when it comes to differences in the activity. This is the weakness of the manuscript.

We would direct reviewer's attention to #1 in the Referee's cross-commenting section above.

Response to Reviewer 2

This is a very nice paper by the Andreu and Torrent groups that report the antimicrobial and heparin-binding of several encrypted peptides. Overall, this study presents an intriguing exploration into the potential dual functionality of glycosaminoglycan (GAG)-binding proteins, specifically heparin-binding proteins (HBPs), in recognizing lipopolysaccharide (LPS) and exhibiting antimicrobial properties. The findings, particularly the identification and characterization of novel encrypted peptides, such as HBP-5, are promising and contribute to our understanding of the intricate interplay between GAG-binding proteins and immunity. The data provided and methodology are thorough and well described. In sum, this is a very nice work. Please see below my minor comments.

Minor comments:

1. Fig. 1 legend does not show antimicrobial activity. Please remove from the figure legend title.

As pointed out by the reviewer, the legend was incorrect and has been corrected accordingly and now reads "Figure 1. Structural and bioinformatics analysis of HBPs".

2. Discussion section: the authors should expand this section a bit to discuss recent work in the encrypted/cryptic peptide area. There are some recent relevant papers published in the past 3 years that should be discussed.

We agree with the reviewer's suggestion to expand the discussion section to address recent work in the field of encrypted/cryptic peptides. We have carefully reviewed the recent literature and added several references in this topic:

Torres MDT, Melo MCR, Flowers L, Crescenzi O, Notomista E, de la Fuente-Nunez C. Mining for encrypted peptide antibiotics in the human proteome. *Nat Biomed Eng.* 2022, **6**:67-75.

Santos MFDS, Freitas CS, Verissimo da Costa GC, Pereira PR, Paschoalin VMF. Identification of Antibacterial Peptide Candidates Encrypted in Stress-Related and Metabolic *Saccharomyces cerevisiae* Proteins. *Pharmaceuticals (Basel)*. 2022, **15**:163.

Boaro A, Ageitos L, Torres MT, Blasco EB, Oztekin S, de la Fuente-Nunez C. Structure-function-guided design of synthetic peptides with anti-infective activity derived from wasp venom. *Cell Rep Phys Sci.* 2023, **4**:101459.

3. References provided are a bit outdated and do not accurately reflect the latest in the field (see comment above).

We thank the reviewer for this comment. Older references were updated as suggested.

4. Gram should be capitalized throughout the text.

Gram has been capitalized as suggested by the reviewer.

5. Can the authors comment on the potential translatability of HBP-5? Please also comment on the potential advantages of having peptides that 1) bind to heparin; and 2) kill bacteria.

We appreciate the reviewer's interest in the potential of HBP-5. Indeed, we believe it has promise for clinical applications due to its unique attributes, but further studies, including in vivo experiments and pharmacokinetic assessments, are needed to fully evaluate its potential. The advantages of peptides that bind to heparin and kill bacteria include targeted delivery or localization of therapeutic agents, enhanced efficacy, and minimized off-target effects. HBP-5's ability to perturb outer membrane LPS, a crucial aspect of its antibacterial activity, makes it a promising approach to combat Gram-negative bacterial infections, which are often challenging to treat. By disrupting the outer membrane integrity, HBP-5 may also enhance the susceptibility of Gram-negative bacteria to other antimicrobial agents or host immune responses, underscoring its translational potential for treating bacterial infections.

6. More details on the computational tools and methods used to mine the peptides are needed.

We have updated the Methods section to provide more details on the computational tools used for defining AMPs. Briefly, from the library of heparin-binding proteins obtained from previous studies [2] and AMP scanning for all these proteins was performed using the AMPA tool. The predicted antibacterial segments were located in the 3D structure of their respective proteins. Then, the CPC' motifs were searched in each segment following the criteria previously reported in [3, 4]. The motif involves two cationic residues (Arg or Lys) and a polar residue (preferentially Asn, Gln, Thr, Tyr or Ser), with fairly conserved distances between the carbons and the side chain center of gravity, defining a clip-like structure where heparin would be lodged. This structural motif is highly conserved and can be found in many proteins with reported heparin binding capacity. Finally, for all these regions, docking with a heparin disaccharide was performed using AutoDock Vina to evaluate the potential binding energy.

Response to Reviewer 3

Summary: This manuscript has identified and investigated antimicrobial peptides from GAG binding proteins. Authors hypothesized that due to physiochemical similarity between GAG and LPS, fragments of GAG binding proteins might exert antimicrobial activity particularly against G- bacteria. Authors have identified few such AMPs that demonstrate LPS binding and displayed antibacterial activity. They have also solved NMR structure of the potent peptide and mode of action.

Major comments: AMPs are promising molecules that can serve as lead for the development of therapeutics against MDR bacteria. In particular, currently therapeutic options to treat MDR Gram negative pathogens are limited. The current study is interesting and provides new non-toxic AMPs. Conclusions drawn from the works are largely valid. However, authors should address following comments:

1. The design and characterization of the peptide YI12WF is not described. Previous studies had shown design of β -boomerang peptides (Bhattacharjya and coworkers) that target LPS.

We thank the reviewer for this comment. YI12WF (YVLWKRKRFIFI-amide) has been previously reported [4, 5] and shown to bind LPS with high affinity. YI12WF also contains a CPC' motif that, if deleted, reduces heparin binding [4]. References have been added in the text.

2. Mutations or substitution of the key residues peptide 5 might improve the novelty of the work.

We thank the reviewer for this comment and agree that targeted substitutions in HBP-5 might shed light on the importance of the CPC' motif. As this point was also raised by reviewer 1, we would direct the reviewer's attention to #2 in the *Referees cross-commenting** section above.

3. How these peptides disrupt LPS permeability is not investigated. As LPS is the major target.

We thank the reviewer for this suggestion and have accordingly evaluated the outer membrane (OM) permeability of the peptides by the 1-N-phenyl-naphthylamine (NPN) assay, a widely used method to assess OM integrity in Gram-negative bacteria. NPN is typically unable to cross the intact outer membrane; however, when the membrane is damaged or disrupted, it can penetrate and interact with lipids and proteins inside the cell, leading to an increase in fluorescence which is directly correlated with the degree of OM permeability and serves as an indicator of membrane damage.

Our results, illustrated in the new Figure 2D, show that all peptides are able to disrupt the OM of Gram-negative bacteria comparably to the LL-37 positive control, except for HBP2. Notably, HBP-5 exhibits the highest activity against OM, consistent with findings elsewhere in the manuscript and altogether confirming the ability of HBPs to bind to and disrupt the LPS structure.

4. Are the D-enantiomers of the peptides active against bacteria?

We tested the antibacterial activity of the D-enantiomer of HBP5 (dHBP-5) and found it to be even higher than that of all-L HBP-5 against both Gram-negative and -positive bacteria, probably due to increased proteolytic stability as found in many AMP studies [6, 7]. As for LPS and heparin affinity, L- and D-HBP-5 behaved similarly (Table R1). As expected, the CD signatures of L- and D-HBP-5 were mirror images (Figure R1). These results suggest that the conformation of the CPC' motif is preserved in dHBP5, in tune with all previous results.

Table R1. Antimicrobial activity of HBP-5 and dHBP-5

Antibacterial Activity						
ID	E. Coli	P. Aeruginosa	A. Baumannii	S. Aureus	E. Faecium	L. monocytognes
HPB-5	0.4	0.8	0.2	6.3	25	1.6
dHBP-5	0.1	0.2	0.2	1.6	0.4	0.2

Binding Affinity		
	LPS (EC ₅₀ , μM)	Heparin (% Elution buffer)
HPB-5	0.9 ± 0.7	98.0
dHBP-5	1.1 ± 0.8	97.2

Figure R1. CD spectra of HBP-5 (red line) and dHBP-5 (green line) in LPS (left panel) and heparin (right panel).

5. 3D structure of peptide 5 is solved in DPC micelle which is a mimic for eukaryotic cells. Authors should attempt to determine structure in LPS as shown in several recent studies with potent AMPs thanatin, MSI etc.

We appreciate the suggestion and have indeed attempted to obtain NMR spectra of HBP-5 in LPS micelles. However, we've been hindered by peptide precipitation and, despite considerable efforts, have not been able to obtain satisfactory results thus far. In contrast, we have succeeded in obtaining CD spectra of HBP5 in LPS micelles, showing an α -helix conformation similar to the one in SDS micelles, hence suggesting similar conformation in both environments.

Minor comments: There are examples of AMPs derived from human proteins. Authors should highlight such works.

Other studies have been cited according to the reviewers' comments:

Malmström E, Mörgelin M, Malmsten M, Johansson L, Norrby-Teglund A, Shannon O, Schmidtchen A, Meijers JC, Herwald H. Protein C inhibitor--a novel antimicrobial agent. *PLoS Pathog.* 2009, **5**:e1000698.

Ishihara, J., Ishihara, A., Fukunaga, K. et al. Laminin heparin-binding peptides bind to several growth factors and enhance diabetic wound healing. *Nat Commun.* 2018, **9**:2163.

Chillakuri Chandramouli R, Jones Céline and Mardon Helen J. Heparin binding domain in vitronectin is required for oligomerization and thus enhances integrin mediated cell adhesion and spreading, *FEBS Letters.* 2010, **584**:3287-91.

Papareddy P, Kasetty G, Kalle M, Bhongir RK, Mörgelin M, Schmidtchen A, Malmsten M. NLF20: an antimicrobial peptide with therapeutic potential against invasive *Pseudomonas aeruginosa* infection. *J Antimicrob Chemother.* 2016 **71**:170-80.

References

1. Papareddy, P., et al., *An antimicrobial helix A-derived peptide of heparin cofactor II blocks endotoxin responses in vivo.* *Biochimica et Biophysica Acta (BBA) - Biomembranes*, 2014. **1838**(5): p. 1225-1234.
2. Ori, A., M.C. Wilkinson, and D.G. Fernig, *A systems biology approach for the investigation of the heparin/heparan sulfate interactome.* *J Biol Chem*, 2011. **286**(22): p. 19892-904.
3. Torrent, M., et al., *The "CPC Clip Motif": A Conserved Structural Signature for Heparin-Binding Proteins.* *PLOS ONE*, 2012. **7**(8): p. e42692.
4. Pulido, D., et al., *Structural similarities in the CPC clip motif explain peptide-binding promiscuity between glycosaminoglycans and lipopolysaccharides.* *J R Soc Interface*, 2017. **14**(136): p. 20170423.
5. Bhunia, A., et al., *Designed beta-boomerang antiendotoxic and antimicrobial peptides: structures and activities in lipopolysaccharide.* *J Biol Chem*, 2009. **284**(33): p. 21991-22004.
6. Varponi, I., et al., *Fighting Pseudomonas aeruginosa Infections: Antibacterial and Antibiofilm Activity of D-Q53 CecB, a Synthetic Analog of a Silkworm Natural Cecropin B Variant.* *Int J Mol Sci*, 2023. **24**(15): p. 12496.
7. Chen, Y., et al., *Comparison of Biophysical and Biologic Properties of α -Helical Enantiomeric Antimicrobial Peptides.* *Chemical Biology & Drug Design*, 2006. **67**(2): p. 162-173.

Dear Dr. Torrent,

Thank you for submitting a manuscript entitled "Mining the heparinome for cryptic antimicrobial peptides that selectively kill gram-negative bacteria" for consideration for publication in Molecular Systems Biology.

Your paper and point-by-point response have now been seen by the Editors of the Journal. We have decided to return it to you with the message that we cannot offer publication of the manuscript in Molecular Systems Biology.

In this study, you inspected all reported heparin-binding proteins (HBPs) to identify potential antimicrobial peptides (AMPs) derived from HBPs, based on using an existing antimicrobial region prediction algorithm. We appreciate that you reported more than 200 potential encrypted peptide antibiotics and further synthesized five peptides which show antimicrobial activity against gram-negative bacteria in vitro. We also acknowledge that your data highlight the importance of the CPC' clip motif for heparin and LPS recognition as well as for the antimicrobial activity.

We recognize that the referees from Review Commons have provided some positive comments on your work while also pointing out its limitations. We also recognize that you can address some of the raised concerns. However, given that previous studies already reported the presence of antimicrobial domain in various HBPs and the CPC motif has been reported in your prior work, we feel that the overall advance is somewhat limited. Taking these into consideration and in line with Referee #1's comment, we think in this case it would be essential to demonstrate the in vivo efficacy of the reported AMPs. As such, we are unfortunately not convinced that the study as it stands provides the degree of biological insight and the level of advance for a broader audience as would be required for publication in Molecular Systems Biology.

I apologize for not being able to bring better news on this occasion but I hope that this early decision will allow you to decide how to proceed with your manuscript without undue delay.

Sincerely,
Jingyi

Jingyi Hou, PhD
Scientific Editor
Molecular Systems Biology

Rev_Com_number: RC-2023-02214

New_manu_number: MSB-2024-12458-T

Corr_author: Torrent

Title: Mining the heparinome for cryptic antimicrobial peptides that selectively kill gram-negative bacteria

Response to Reviewer 1

Major comments

1. Even though the Authors propose here that CPC' clip motif is needed for antimicrobial activity. However, various studies have demonstrated that the mere presence of cationic amino or hydrophobic amino acids does not give the activity, the location of these amino acids at the strategic position is critically needed. The major issue in this work, the authors have not presented, whether there was a single CPC motif or multiple in the 5 peptides they have synthesized. Further, they need to demonstrate how are the charged and hydrophobic amino acids distributed in the peptides. these things will clearly explain the difference in the activity as well spectrum of the peptides. The authors should make an extra figure or add information highlighting this unique characteristic for better understanding to the reader.

We thank the reviewer for his/her comments and suggestions. We concur that the distribution of amino acids is crucial for the antimicrobial activity of the peptides and their ability to bind heparin. We also agree with the suggestion of illustrating the location of the CPC' motifs of HBPs in the context of the parental proteins and have accordingly done so in the new Supplementary Figure 1. In all cases, only one CPC' motif was identified in the antimicrobial region, as highlighted in the figure, and the inter-residue distances measured are consistent with the CPC' motif definition. Thus, we demonstrate that a CPC' motif exists in all five HBPs, which explains how they recognize and bind heparin.

To illustrate the distribution of charged and hydrophobic amino acids in HBPs, we have also prepared new Supplementary Figure 2, displaying electrostatic potentials in the predicted HBP structures, and showing how the distribution of charged residues creates hydrophobic and cationic patches on the surface of the peptides. Our analysis reveals cationic patches to be surrounded by hydrophobic residues, which may explain the ability of the peptides to disrupt membranes and exert antimicrobial activity.

2. It is strange to observe that there are quite a number of reports showing that the peptides derived from the Herprin binding proteins have antimicrobial activity, but no one has reported their efficacy in the in vivo mouse model. if possible, the authors could add their observations if in vivo studies were done. or as a future line of study.

We thank the reviewer for his/her comment on the observation of antimicrobial activity in peptides derived from heparin-binding proteins. Indeed, a few such studies have appeared in the literature, some with moderate success [1]. Often, such peptides fail to demonstrate significant in vivo activity due to the need for optimization to enhance serum stability. To evaluate the potential of HBP-5 in vivo, we performed in vitro protease digestion assays with human serum for both HBP-5 and its D-enantiomeric version, dHBP-5. The results revealed that HBP-5 exhibited low to moderate stability in serum, while dHBP-5 demonstrated a significantly prolonged half-life exceeding 6 hours. The enhanced stability of dHBP-5 is attributed to its enantiomeric configuration, which protects it from protease degradation. Importantly, dHBP-5 maintained low cytotoxicity and hemolytic activity, making it a promising candidate for in vivo testing.

Additionally, we identified a stable fragment of HBP-5, named HBP-5[1-20], spanning the first 20 residues and retaining the CPC' motif. This fragment exhibited enhanced serum stability than the original peptide while preserving equivalent antimicrobial activity, cytotoxicity, and hemolysis. Based on these findings, HBP-5[1-20] was also deemed a suitable candidate for in vivo evaluation. Consequently, we proceeded with in vivo studies using three peptides: the original HBP-5, dHBP-5, and HBP-5[1-20].

The in vivo antimicrobial efficacy of these peptides was assessed in a murine model of sepsis induced by *Acinetobacter baumannii*. Results showed a substantial reduction in colony-forming units (CFUs) across all organs, ranging from 2 to 5 orders of magnitude, depending on the organ and sex of the animals (See Figure R1 below, corresponding to Figure 5 in the manuscript). Notably, all peptides outperformed LL-37, which served as a positive control. The HBP-5[1-20] fragment exhibited particularly strong activity in females, achieving CFU reductions of 4 to 5 orders of magnitude across all analyzed organs (Figure 5B).

In this context, we believe our results are encouraging and will spur further efforts, specifically by providing a rationale on how to identify CPC' motifs hence heparin-binding regions in protein sequences.

Figure R1. Anti-infective activity of peptides in animal model. (A) Schematic of the sepsis mouse model used to assess the anti-infective activity of selected antimicrobials peptides (n = 6) against *A. baumannii* ATCC 15308. (B) Average CFU/mg in mice organs after 18 hours of treatment. Statistical significance compared to vehicle was determined using two-way ANOVA followed by Dunnett' s test (denoted as **** p < 0.0001).

Referees cross-commenting*

Minor comments

1a. The presence of Cryptic antimicrobial domain in various heparin-binding proteins like laminin isoforms, von Willebrand factor, vitronectin, protein C inhibitor, matrix glycoproteins thrombospondin, proline arginine-rich end leucine-rich repeat protein and fibronectin, have been reported previous. It is not clear why the authors did not refer to that work. The authors should refer to the works. (same as reviewer 3)

We were aware of other prior studies on heparin-binding proteins and did indeed cite some of them, though not exhaustively for conciseness' sake. However, as encouraged by reviewers 1 and 3 we have cited the following studies:

Malmström E, Mörgelin M, Malmsten M, Johansson L, Norrby-Teglund A, Shannon O, Schmidtchen A, Meijers JC, Herwald H. Protein C inhibitor--a novel antimicrobial agent. *PLoS Pathog.* 2009 Dec;5(12):e1000698. doi: 10.1371/journal.ppat.1000698. Epub 2009 Dec 18. PMID: 20019810; PMCID: PMC2788422.

Ishihara, J., Ishihara, A., Fukunaga, K. et al. Laminin heparin-binding peptides bind to several growth factors and enhance diabetic wound healing. *Nat Commun* 9, 2163 (2018). <https://doi.org/10.1038/s41467-018-04525-w>

Chillakuri Chandramouli R, Jones Céline and Mardon Helen J(2010), Heparin binding domain in vitronectin is required for oligomerization and thus enhances integrin mediated cell adhesion and spreading, *FEBS Letters*, 584, doi: 10.1016/j.febslet.2010.06.023

Papareddy P, Kasetty G, Kalle M, Bhongir RK, Mörgelin M, Schmidtchen A, Malmsten M. NLF20: an antimicrobial peptide with therapeutic potential against invasive *Pseudomonas aeruginosa* infection. *J Antimicrob Chemother.* 2016 Jan;71(1):170-80. doi: 10.1093/jac/dkv322. Epub 2015 Oct 26. PMID: 26503666.

1b. All the earlier studies related to the antimicrobial activity of the peptides derived from the Heparin-binding protein reported a consensus Cardin and Weintraub motifs i.e, XBBBXXBX or XBBXBX, where X represents hydrophobic or uncharged amino acids, and B represents basic amino acids. However, in this work, the researchers report about the presence of the new CPC motif. So, this is unique and a novelty in the study.

We thank the reviewers for these observations. Indeed, our quest to unveil CPC' motifs in antimicrobial regions of heparin-binding proteins is the key point of our investigation, and what distinguishes it from previous studies on consensus motifs such as XBBBXXBX or XBBXBX. We believe our definition of CPC' motifs in simple, structure-based, and experimentally verifiable terms is not only a significant departure but also a step forward from earlier views, highlighting the importance of a structural perspective in defining heparin-binding regions. In point of fact, we show that our peptides, even without consensus Cardin-Weintraub motifs, bind heparin with high affinity. The presence of the CPC' motif is crucial for such binding, as well as for LPS binding, and the new experiments performed at editor/reviewer's request, where the CPC motif in HBP5 is abolished, with predictable impact, fully support our view, see new section "Insights into the CPC' motif of HBP-5 and its implication on the antibacterial mechanism" and new Table 3 in the revised manuscript.

2. Even though the researchers report on the role of the CPC motif in the antimicrobial activity and binding to the heparin, the authors did not show any data or draw the conclusions related to the CPC domain when it comes to differences in the activity. this is the weakness of the manuscript. (same as reviewer 2)

We welcome the reviewer's observation. To address it, we made and tested three HBP-5 mutants aimed at showing how alterations in the CPC' motif might influence interaction with heparin and LPS, as well as antimicrobial properties. The first two mutants involved replacing positively charged R10 and R14 residues with glutamine, similar in size and polarity but uncharged. As shown in the new section "Insights into the CPC' motif of HBP-5 and its implication on the antibacterial mechanism" and on the new Table 3 of the revised manuscript, the changes reduced heparin binding, i.e., shorter retention times on affinity chromatography, as well as LPS binding, i.e., a decrease in EC₅₀ in the cadaverine assay (Table 3). The modifications had a lesser impact on antimicrobial activity, most likely due to the low resolution of MIC assays.

In a further step to assess the effect of the CPC' motif on antimicrobial activity, we deleted it in full by replacing residues H9, R10 and R14 of HBP-5 by alanine. As expected, this Δ CPC' peptide showed a sharp reduction in both heparin and LPS binding (Table 3) and, most importantly, a significant and asymmetric change in antimicrobial activity, with substantial impact on Gram-negatives yet practically no effect on Gram-positives, suggesting that LPS plays a key role in this selective response. Altogether, these observations align with our hypothesis that heparin-binding proteins might exploit their intrinsic affinity for heparin as an opportunity to developing antimicrobial properties by leveraging structural similarities between glycosaminoglycans and LPS.

3. It is strange to observe that there are quite a number of reports showing that the peptides derived from the Heparin (*sic*) binding proteins have antimicrobial activity, but no one has reported their efficacy in the in vivo mouse model. if possible, the authors could add their observations if in vivo studies were done. or as a future line of study. (Same as reviewer 2)

We would kindly direct attention to #2 in the response to reviewer 1 above.

4. There are more than 20 different AMP databases or prediction software. however, not all of them are 100 % current, their success rate varies from 30-50% only. It needs to be investigated if adding this search in the hit peptides might increase the success rate of the extra in silico-based AMPs prediction software.

If we understand the question correctly, the reviewer wonders whether including a CPC' motif predictor would increase the accuracy of AMP search algorithms. In our view, this strategy has two main limitations to be considered: (i) locating a CPC' motif in a peptide sequence typically requires a known 3D structure. Unfortunately, this is not always the case, and for proteins lacking reliable 3D data it can be a challenging and resource-intensive process; (ii) while CPC' motifs may predispose proteins to evolve antimicrobial properties, it is unclear if this is a required feature for all AMPs. Imposing the presence of a CPC' motif may not be applicable to all AMPs, although it might help identifying peptides with specific activity against gram-negative strains.

In summary, while the query of including a CPC' motif search tool in AMP predictors is intriguing and worthy of exploration for its potential bearing on antimicrobial research, it is technically complicated and beyond the scope of our manuscript.

Reviewer #1 (Significance (Required)):

All the earlier studies related to the antimicrobial activity of the peptides derived from the Heparin-binding protein reported a consensus Cardin and Weintraub motifs i.e, XBBBXXBX or XBBXBX, where X represents hydrophobic or uncharged amino acids, and B represents basic amino acids. However, in this work, the researchers report about the presence of the new CPC motif. So this is unique and a novelty in the study.

Even though the researchers report on the role of the CPC motif in the antimicrobial activity and binding to the heparin, the authors did not show any data or draw conclusions related to the CPC domain when it comes to differences in the activity. This is the weakness of the manuscript.

We would direct reviewer's attention to #1b and #2 in the Referee's cross-commenting section above.

Response to Reviewer 2

Minor comments:

1. Fig. 1 legend does not show antimicrobial activity. Please remove from the figure legend title.

As pointed out by the reviewer, the legend was incorrect and has been corrected accordingly and now reads "Figure 1. Structural and bioinformatics analysis of HBPs".

2. Discussion section: the authors should expand this section a bit to discuss recent work in the encrypted/cryptic peptide area. There are some recent relevant papers published in the past 3 years that should be discussed.

We agree with the reviewer's suggestion to expand the discussion section to address recent work in the field of encrypted/cryptic peptides. We have carefully reviewed the recent literature and added several references in this topic:

Torres MDT, Melo MCR, Flowers L, Crescenzi O, Notomista E, de la Fuente-Nunez C. Mining for encrypted peptide antibiotics in the human proteome. Nat Biomed Eng. 2022 Jan;6(1):67-75. doi: 10.1038/s41551-021-00801-1. Epub 2021 Nov 4. Erratum in: Nat Biomed Eng. 2022 Dec;6(12):1451. PMID: 34737399.

*Santos MFDS, Freitas CS, Verissimo da Costa GC, Pereira PR, Paschoalin VMF. Identification of Antibacterial Peptide Candidates Encrypted in Stress-Related and Metabolic *Saccharomyces cerevisiae* Proteins. Pharmaceuticals (Basel). 2022 Jan 28;15(2):163. doi: 10.3390/ph15020163. PMID: 35215278; PMCID: PMC8877035.*

Boaro A, Ageitos L, Torres MT, Blasco EB, Oztekin S, de la Fuente-Nunez C. Structure-function-guided design of synthetic peptides with anti-infective activity derived from wasp venom. Cell Rep Phys Sci. 2023 Jul 19;4(7):101459. doi: 10.1016/j.xcrp.2023.101459. PMID: 38239869; PMCID: PMC10795512.

3. References provided are a bit outdated and do not accurately reflect the latest in the field (see comment above).

We thank the reviewer for this comment. Older references were updated as suggested.

4. Gram should be capitalized throughout the text.

Gram has been capitalized as suggested by the reviewer.

5. Can the authors comment on the potential translatability of HBP-5? Please also comment on the potential advantages of having peptides that 1) bind to heparin; and 2) kill bacteria.

We appreciate the reviewer's interest in the potential of HBP-5. Based on the in vivo results obtained, we believe it has promise for clinical applications due to its unique attributes, but further studies, including pharmacokinetic and pharmacodynamic assessments. The advantages of peptides that bind to heparin and kill bacteria include targeted delivery or localization of therapeutic agents, enhanced efficacy, and minimized off-target effects. HBP-5's ability to perturb outer membrane LPS, a crucial aspect of its antibacterial activity, makes it a promising approach to combat Gram-negative bacterial infections, which are often challenging to treat. By disrupting

the outer membrane integrity, HBP-5 may also enhance the susceptibility of Gram-negative bacteria to other antimicrobial agents or host immune responses, underscoring its translational potential for treating bacterial infections.

6. More details on the computational tools and methods used to mine the peptides are needed.

We have updated the Methods section to provide more details on the computational tools used for defining AMPs. Briefly, from the library of heparin-binding proteins obtained from previous studies [2] and AMP scanning for all these proteins was performed using the AMPA tool. The predicted antibacterial segments were located in the 3D structure of their respective proteins. Then, the CPC' motifs were searched in each segment following the criteria previously reported in [3, 4]. The motif involves two cationic residues (Arg or Lys) and a polar residue (preferentially Asn, Gln, Thr, Tyr or Ser), with fairly conserved distances between the carbons and the side chain center of gravity, defining a clip-like structure where heparin would be lodged. This structural motif is highly conserved and can be found in many proteins with reported heparin binding capacity. Finally, for all these regions, docking with a heparin disaccharide was performed using AutoDock Vina to evaluate the potential binding energy.

Response to Reviewer 3

Major comments: AMPs are promising molecules that can serve as lead for the development of therapeutics against MDR bacteria. In particular, currently therapeutic options to treat MDR Gram negative pathogens are limited. The current study is interesting and provides new non-toxic AMPs. Conclusions drawn from the works are largely valid. However, authors should address following comments:

1. The design and characterization of the peptide YI12WF is not described. Previous studies had shown design of β -boomerang peptides (Bhattacharjya and coworkers) that target LPS.

We thank the reviewer for this comment. YI12WF (YVLWKRKRKFIFI-amide) has been previously reported [4, 5] and shown to bind LPS with high affinity. YI12WF also contains a CPC' motif that, if deleted, reduces heparin binding [4]. Peptide YI12WF has been cited in the Introduction (page 2) and Results (page 10) and references have been included in the main manuscript.

2. Mutations or substitution of the key residues peptide 5 might improve the novelty of the work.

We thank the reviewer for this comment and agree that targeted substitutions in HBP-5 might shed light on the importance of the CPC' motif. As this point was also raised by reviewer 1, we would direct the reviewer's attention to #2 in the *Referees cross-commenting** section above.

3. How these peptides disrupt LPS permeability is not investigated. As LPS is the major target.

We thank the reviewer for this suggestion and have accordingly evaluated the outer membrane (OM) permeability of the peptides by the 1-N-phenyl-naphthylamine (NPN) assay, a widely used

method to assess OM integrity in Gram-negative bacteria. NPN is typically unable to cross the intact outer membrane; however, when the membrane is damaged or disrupted, it can penetrate and interact with lipids and proteins inside the cell, leading to an increase in fluorescence which is directly correlated with the degree of OM permeability and serves as an indicator of membrane damage.

Our results, illustrated in the new Figure 2D, show that all peptides are able to disrupt the OM of Gram-negative bacteria comparably to the LL-37 positive control, except for HBP2. Notably, HBP-5 exhibits the highest activity against OM, consistent with findings elsewhere in the manuscript and altogether confirming the ability of HBPs to bind to and disrupt the LPS structure.

4. Are the D-enantiomers of the peptides active against bacteria?

We tested the antibacterial activity of the D-enantiomer of HBP5 (dHBP-5) and found it to be even higher than that of all-L HBP-5 against both Gram-negative and -positive bacteria, probably due to increased proteolytic stability as found in many AMP studies [6, 7]. As for LPS and heparin affinity, L- and D-HBP-5 behaved similarly (Table R1). As expected, the CD signatures of L- and D-HBP-5 were mirror images (Figure R2). These results suggest that the conformation of the CPC' motif is preserved in dHBP5, in tune with all previous results. Additionally, results obtained in vivo in a mouse model of sepsis further indicate that the activity of dHBP5 is preserved (please see #2 in the response to reviewer 1 above).

Table R1. Antimicrobial activity of HBP-5 and dHBP-5

Antibacterial Activity						
ID	E. Coli	P. Aeruginosa	A. Baumannii	S. Aureus	E. Faecium	L. monocytognes
HPB-5	0.4	0.8	0.2	6.3	3.1	1.6
dHBP-5	0.1	0.2	0.2	1.6	1.6	0.8

Binding Affinity		
	LPS (EC ₅₀ , μM)	Heparin (% Elution buffer)
HPB-5	0.9 ± 0.7	98.0
dHBP-5	1.1 ± 0.8	97.2

Figure R2. CD spectra of HBP-5 (red line) and dHBP-5 (green line) in all conditions tested.

5. 3D structure of peptide 5 is solved in DPC micelle which is a mimic for eukaryotic cells. Authors should attempt to determine structure in LPS as shown in several recent studies with potent AMPs thanatin, MSI etc.

We appreciate the suggestion and have indeed attempted to obtain NMR spectra of HBP-5 in LPS micelles. However, we've been hindered by peptide precipitation and, despite considerable efforts, have not been able to obtain satisfactory results thus far. In contrast, we have succeeded in obtaining CD spectra of HBP5 in LPS micelles, showing an α -helix conformation similar to the one in SDS micelles, hence suggesting similar conformation in both environments.

Minor comments: There are examples of AMPs derived from human proteins. Authors should highlight such works.

Other studies have been cited according to the reviewers' comments:

Malmström E, Mörgelin M, Malmsten M, Johansson L, Norrby-Teglund A, Shannon O, Schmidtchen A, Meijers JC, Herwald H. *Protein C inhibitor--a novel antimicrobial agent. PLoS Pathog.* 2009 Dec;5(12):e1000698. doi: 10.1371/journal.ppat.1000698. Epub 2009 Dec 18. PMID: 20019810; PMCID: PMC2788422.

Ishihara, J., Ishihara, A., Fukunaga, K. et al. *Laminin heparin-binding peptides bind to several growth factors and enhance diabetic wound healing. Nat Commun* 9, 2163 (2018). <https://doi.org/10.1038/s41467-018-04525-w>

Chillakuri Chandramouli R, Jones Céline and Mardon Helen J(2010), *Heparin binding domain in vitronectin is required for oligomerization and thus enhances integrin mediated cell adhesion and spreading, FEBS Letters*, 584, doi: 10.1016/j.febslet.2010.06.023

Papareddy P, Kasetty G, Kalle M, Bhongir RK, Mörgelin M, Schmidtchen A, Malmsten M. *NLF20: an antimicrobial peptide with therapeutic potential against invasive Pseudomonas aeruginosa infection. J Antimicrob Chemother.* 2016 Jan;71(1):170-80. doi: 10.1093/jac/dkv322. Epub 2015 Oct 26. PMID: 26503666.

References

1. Papareddy, P., et al., *An antimicrobial helix A-derived peptide of heparin cofactor II blocks endotoxin responses in vivo. Biochimica et Biophysica Acta (BBA) - Biomembranes*, 2014. **1838**(5): p. 1225-1234.
2. Ori, A., M.C. Wilkinson, and D.G. Fernig, *A systems biology approach for the investigation of the heparin/heparan sulfate interactome. J Biol Chem*, 2011. **286**(22): p. 19892-904.
3. Torrent, M., et al., *The "CPC Clip Motif": A Conserved Structural Signature for Heparin-Binding Proteins. PLOS ONE*, 2012. **7**(8): p. e42692.
4. Pulido, D., et al., *Structural similarities in the CPC clip motif explain peptide-binding promiscuity between glycosaminoglycans and lipopolysaccharides. J R Soc Interface*, 2017. **14**(136).
5. Bhunia, A., et al., *Designed beta-boomerang antiendotoxic and antimicrobial peptides: structures and activities in lipopolysaccharide. J Biol Chem*, 2009. **284**(33): p. 21991-22004.
6. Varponi, I., et al., *Fighting Pseudomonas aeruginosa Infections: Antibacterial and Antibiofilm Activity of D-Q53 CecB, a Synthetic Analog of a Silkworm Natural Cecropin B Variant. Int J Mol Sci*, 2023. **24**(15).
7. Chen, Y., et al., *Comparison of Biophysical and Biologic Properties of α -Helical Enantiomeric Antimicrobial Peptides. Chemical Biology & Drug Design*, 2006. **67**(2): p. 162-173.

4th Feb 2025

Manuscript Number: MSB-2024-12458R-Q

Title: Mining the heparinome for cryptic antimicrobial peptides that selectively kill Gramnegative bacteria

Author: Daniel Sandín García

Roberto Bello-Madruga

Javier Valle

Jordi Gomez Borrego

Laura Comas

María Nieves Larrosa Escartín

Juan José López González

María Angeles Jiménez

David Andreu

Marc Torrent

Dear Marc,

Thank you for submitting your revised manuscript. We have now heard back from the two reviewers who agreed to evaluate your study. As you will see below, the reviewers are satisfied with the performed revisions and support publication. Before we can formally accept the manuscript for publication, we would ask you to address some remaining editorial-level issues listed below.

1. Please provide a .docx formatted version of the manuscript text (including legends for main figures, EV figures and tables). Please make sure that the changes are highlighted to be clearly visible.
2. Please provide individual production quality figure files as .eps, .tif, .jpg (one file per figure). Figures should be removed from the manuscript file.
3. At EMBO Press we ask authors to provide source data for the main figures. Our source data coordinator will contact you to discuss which figure panels we would need source data for and will also provide you with helpful tips on how to upload and organize the files.
4. Reference format: citations should be listed in an alphabetical order and list 10 co-authors of a paper before adding et al.
5. Please ensure that the funding information is consistent between the manuscript and the online submission system.
 - Missing in the manuscript: European Society of Clinical Microbiology and Infectious Diseases (ESCMID) ESCMID2022;
 - Missing in the online submission system: the Spanish Ministry of Science and Innovation with funds from the European Union NextGenerationEU, from the Recovery, Transformation and Resilience Plan (PRTR-C17.11) and from the Autonomous Community of Catalonia within the framework of the Biotechnology Plan Applied to Health; the MCI María de Maeztu network of Units of Excellence; pre-doctoral a FPI scholarship (PRE2018-083243 and PRE2021-097678 from the Spanish Ministerio de Ciencia e Innovación; all missing funders should be added with the option "More Funders", not copied in the Comments box.
6. Appendix:
 - the title page should contain "Appendix for + manuscript title" and a Table of Content with the page numbers for the listed items.
 - nomenclature should be Appendix Figure Sx and Appendix Table Sx throughout the manuscript and Appendix PDF.
 - highlights should be removed from text, as Appendix PDF will not be typeset.
7. Author contribution should be entered in the online submission system for each author.
8. "Competing interests" should be renamed to "Disclosure and Competing interests Statement" .
9. Please add a "Data availability" section. Since this study does not generate large-scale datasets, please only include the following sentence in this section- "This study includes no data deposited in external repositories".
10. Please address the following issues in Figure Legends :
 - Please note that the exact p values are not provided in the legends of figures 5B
 - Please note that information related to n is missing in the legend of figure 1D.
 - Please note that the error bars are not defined in the legend of figure 1D.
11. Section order should be corrected: Title page - Abstract & Keywords - Introduction - Results - Discussion - Methods - Data Availability - Acknowledgements - Disclosure and Competing Interests Statement - References - Figure Legends - Table(s) -

Expanded View Figure Legends.

12. All Materials and Methods need to be described in the main text using our 'Structured Methods' format. According to this format, the Methods section includes a Reagents and Tools Table (listing key reagents, experimental models, software and relevant equipment and including their sources and relevant identifiers) followed by a Methods and Protocols section describing the methods, ideally using a step-by-step protocol format. The aim is to facilitate adoption of the methodologies across labs. Please use the heading "Methods".

Please download and fill our Reagents and Tools Table template (.docx), which you can find in our author guidelines: <https://www.embopress.org/page/journal/17444292/authorguide#structuredmethods>.

An example of a Method paper with Structured Methods can be found here: <https://www.embopress.org/doi/10.15252/msb.20178071>.

13. Please provide a "standfirst text" summarizing the study in one or two sentences (approximately 250 characters, including space), three to four "bullet points" highlighting the main findings and a "synopsis image" (550px width and 400-600 px height, PNG format) to highlight the paper on our homepage.

Here are a couple of examples:

<https://www.embopress.org/doi/10.15252/msb.20199356>

<https://www.embopress.org/doi/10.15252/msb.20209475>

<https://www.embopress.org/doi/10.15252/msb.209495>

When you resubmit your manuscript, please download our CHECKLIST (<https://www.embopress.org/pb-assets/embosite/EMBO%20Press%20Author%20Checklist-1642513524327.xlsx>) and include the completed form in your submission.

Please note that the Author Checklist will be published alongside the paper as part of the transparent process (<https://www.embopress.org/page/journal/17444292/authorguide#transparentprocess>).

Click on the link below to submit your revised paper.

Kind regards,
Jingyi

Jingyi Hou, PhD
Senior Editor
Molecular Systems Biology

Reviewer #1:

The revised version of the manuscript is well suitable for publication. New data have significantly strengthen the manuscript.

Reviewer #2:

The authors have adequately addressed the reviewer's comments. The manuscript can be accepted in its current form.

All editorial and formatting issues were resolved by the authors.

7th May 2025

Manuscript number: MSB-2024-12458RR

Title: Mining the heparinome for cryptic antimicrobial peptides that selectively kill Gram-negative bacteria

Dear Marc,

Thank you again for sending us your revised manuscript. We are now satisfied with the modifications made and I am pleased to inform you that your paper has been accepted for publication.

Sincerely,
Jingyi

Jingyi Hou, PhD
Senior Editor
Molecular Systems Biology
